# Anticancer Nanoparticle Carriers of the Proapoptotic Protein Cytochrome *c*

**DOI:** 10.3390/pharmaceutics17030305

**Published:** 2025-02-26

**Authors:** Alexandar M. Zhivkov, Svetlana H. Hristova, Trifon T. Popov

**Affiliations:** 1Scientific Research Center, “St. Kliment Ohridski” Sofia University, 8 Dragan Tsankov Blvd., 1164 Sofia, Bulgaria; 2Department of Medical Physics and Biophysics, Medical Faculty, Medical University—Sofia, Zdrave Str. 2, 1431 Sofia, Bulgaria; 3Faculty of Physics, Sofia University, 5 James Bourchier Blvd., 1164 Sofia, Bulgaria; 4Medical Faculty, Medical University—Sofia, Zdrave Str. 2, 1431 Sofia, Bulgaria

**Keywords:** cytochrome *c*, cancer, apoptosis, cytotoxicity, cancer cell cultures, nanoparticles

## Abstract

This review discusses the literature data on the synthesis, physicochemical properties, and cytotoxicity of composite nanoparticles bearing the mitochondrial protein cytochrome *c* (cytC), which can act as a proapoptotic mediator in addition to its main function as an electron carrier in the electron transport chain. The introduction of exogenous cytC via absorption of carrier particles, the phagocytosis of colloid particles of submicrometric size, or the receptor-mediated endocytosis of nanoparticles in cancer cells, initiates the process of apoptosis—a multistage cascade of biochemical reactions leading to complete destruction of the cells. CytC–carrier composite particles have the potential for use in the treatment of neoplasms with superficial localization: skin, mouth, stomach, colon, etc. This approach can solve the two main problems of anticancer therapy: selectivity and non-toxicity. Selectivity is based on the incapability of the normal cell to absorb (nano)particles, except for the cells of the immune system. The use of cytC as a protein that normally functions in mitochondria is harmless for the macroorganism. In this review, the factors limiting cytotoxicity and the ways to increase it are discussed from the point of view of the physicochemical properties of the cytC–carrier particles. The different techniques used for the preparation of cytC-bearing colloids and nanoparticles are discussed. Articles reporting the achievement of high cytotoxicity with each of the techniques are critically analyzed.

## 1. Introduction

### 1.1. Anticancer Strategies

The main traditional methods of neoplasm management are surgery, radiotherapy, and chemotherapy [1]. In conventional chemotherapy, different agents are applied, such as antimetabolites (5-fluorouracil, 6-mercaptopurine, methotrexate, etc.), alkylating agents (cyclophosphamide), platinum compounds (cisplatin, carboplatin, and others), microtubule inhibitors (paclitaxel, docetaxel, vinblastine, vincristine, etc.), and topoisomerase I inhibitors (camptothecin) [2]. A major disadvantage of anticancer chemotherapeutics is that healthy cells are attacked along with cancer cells; for example, blocking DNA synthesis affects all dividing cells, including epithelial, hematopoietic, and immune ones [3]. The damage to the immune system (immunosuppression) is particularly dangerous: after a course of chemotherapy, a patient may die from a viral or bacterial infection, especially considering the increasing number of pathogenic bacteria strains that are resistant to current antibiotics [4]. In addition, the lack of selectivity requires high doses of chemotherapeutic treatment in order to achieve an adequate concentration of the drug in the cancer tissue, which is associated with numerous side effects: hepatotoxicity, nephrotoxicity, depression of hematopoiesis (resulting in anemia, leukopenia, and thrombocytopenia), cardiotoxicity, and also nausea and vomitus [5]. This makes it necessary to look for ways to act only on cancer cells without affecting healthy ones, i.e., the selectivity of action must be increased. The second disadvantage of anticancer chemotherapy is that the killing of cancer cells occurs by the mechanism of necrosis, in which the destruction of neoplastic tissue is accompanied by the release of numerous substances from the decaying cells that are harmful to the macroorganism [6].

In order to overcome the drawbacks of conventional cancer treatment, different drug delivery systems (DDSs), such as nanoparticles, are developed [7,8]. In recent years, in order to increase the selectivity of nanoparticles, various ways of targeting using smart nanoparticles have been developed [9]. Some smart nanoparticles release the carried chemotherapeutic load under the action of triggers such as pH change (the tumor tissue is slightly acidified because of lactate generation), enzymes (some enzymes such as metalproteinases are highly expressed in the tumor cells), reactive oxygen species, temperature, magnetic field, ultrasound, electric field, etc. [10,11,12]. Additionally, the surface of the nanoparticles can be modified, and they can bear specific small molecules and proteins that can bind selectively to overexpressed receptors on the tumor cell membrane [9]. For instance, they are composed of smart nanoparticles that carry folic and hyaluronic acids, transferrin, and monoclonal antibodies against specific cancer receptors such as vascular endothelial growth factor, epidermal growth factor, etc. [13,14]. In addition to triggering apoptosis in cancer cells [15], other approaches to improve anticancer chemotherapy are the reduction in the drug efflux by the blockage of transporter proteins [16] and the inhibition of hypoxia-inducing proteins such as factor 1α (HIF-1α) [17].

### 1.2. Apoptosis in Cancer Cells

Apoptosis (programmed cell death) is a multi-cascade biochemical process leading to cell death [18]. It is characterized by specific morphological and biochemical changes in the cells, including cell shrinkage, condensation and fragmentation of the nucleus, dynamic formation of membrane vesicles, DNA fragmentation, and loss of adhesion to neighboring cells and extracellular matrix [19]. Finally, apoptotic bodies are formed. It is found that apoptotic bodies derived from 3D-cultured stem cells stimulate angiogenesis and proliferation [20]. Apoptosis works as a cell suicide program, causing minimal damage to surrounding tissues (unlike the toxic process of necrosis), and is an important process for tissue formation and maintaining tissue homeostasis [21,22]. The apoptosis deregulation leads to the development of autoimmune diseases, neurodegenerative damage, cancer, and other issues [23,24].

There are two main pathways of apoptosis—extrinsic and intrinsic. These begin after different apoptotic-induced signals and involve different caspase proteolytic enzymes (Figure 1) [25]. Apoptosis in cancer cells has specific differences. In some cancer cells, the expression of death receptors (such as TNFR1 and FasR) and the Fas-associated death domain (FADD) is deregulated [26]. The concentration of external death signals such as tumor necrosis factor (TNF) and Fas ligand (FasL) can also be reduced [27]. The concentration and activity of the major enzymes involved in both pathway caspases decreased in tumor cells [28]. Additionally, the equilibrium between anti-apoptotic (Bcl-2 for instance) and proapoptotic (Bax and Bak) proteins is disturbed in cancer cells: the concentration of proapoptotic ones is reduced, while the expression of anti-apoptotic proteins is increased [29]. Moreover, the expression of inhibitors of apoptosis proteins (IAPs) is upregulated and their level is increased [30]. Additionally, the release of cytochrome *c* from the mitochondria, which activates the caspase cascade after the formation of cytC-Apaf-1 complex with the cytoplasmic protein apoptotic protease-activating factor (Apaf-1), is blocked in cancer cells because of the hyperpolarization of the internal mitochondrial membrane [31].

Cytochrome *c* (cytC) is a small (104 amino acid residues, 12.4 kg/mol molecular mass 3 nm in size) water-soluble globular protein, whose polypeptide chain is coiled around the heme cycle and covalently bound to it, due to which the cytC globule obtains extreme pH and temperature 3D stability [32]. Normally, cytC is located in the mitochondria of the eukaryotic cells, where it works as an Fe^2+^/Fe^3+^ electron carrier between cytochrome *c*_1_ and cytochrome *c*_a_ of the electron transport protein complex of the internal mitochondrial membrane, to which cytC is electrostatically associated. CytC is encoded by nuclear DNA, so it is synthesized in the cytoplasm as a precursor polypeptide (apocytochrome *c*), which penetrates the gap between the inner and outer mitochondrial membranes, where it binds to the heme molecule, forming holocytochrome *c*, and thus cannot exit back into the cytoplasm [33,34].

### 1.3. Apoptosis Targeting in Tumor Cells

Since both intrinsic and extrinsic apoptotic pathways are suppressed in cancer cells, different approaches have been developed to activate apoptosis (Figure 2) [35]. One approach is based on restoring the disturbed balance between proapoptotic and anti-apoptotic proteins [36]. This can be achieved with the use of agents that block Bcl-2 proteins such as sodium butyrate, etc. Another way to enhance apoptosis in cancer cells is via activators of caspases (apoptin, for instance) [37]. The recovery of the tumor suppressor p53 via gene therapy is also a possible approach [38]. Additionally, the apoptosis threshold can be lowered by blockers of inhibitors of apoptosis proteins (IAPs), which can block either the X-linked inhibitor of apoptosis protein (XIAP) or survivin [39]. Agents that generate reactive oxygen species (ROS), such as hydrogen peroxide, singlet oxygen, superoxide anions, and hydroxyl radicals, can induce oxygen stress in cancer cells and lead to apoptosis [40]. Such a ROS generator is α-tocopheryl succinate [41]. The introduction of exogenous cytC is also a promising approach for targeting apoptosis in cancer cells [42].

In cancer cells, apoptosis can be initiated by the introduction of exogenous cytC into them. In their cytoplasm, cytC globules electrostatically bind to the large protein Apaf-1 (apoptotic protease-activating factor 1), which is found as an inactive monomer. CytC binding to Apaf-1 results in a transition of the last from a closed to an open conformation, releasing its oligomerizating domain [43,44]. This leads to the formation of a heptameric complex of seven molecules of Apaf-1 and seven molecules of cytC, called apoptosome. This huge cytC-Apaf-1 protein complex then activates procaspase-9. Functionally active caspase-9 activates effector caspase-3, and this proteolytic enzyme triggers the cell destruction processes, activating a cascade of other enzymes [45,46,47] (Figure 1).

CytC in monomeric form cannot be introduced from aqueous solution into cancer cells because its water-soluble 3 nm globules cannot pass through the hydrophobic bilayer of the cytoplasmic membrane. One way to introduce such exogenous cytC into the cytoplasm of cells is by microinjection, which leads to rapid apoptosis activation [48,49]. Microinjection is a suitable method for laboratory investigation of the mechanisms and kinetics of apoptosis in a single cancer cell [50], but this technique does not allow simultaneous treatment of multiple cells and is not applicable to the clinical treatment of neoplasms. Another method for the selective delivery of cytC into cancer cells is the use of particle carriers of cytC (Figure 3) [51,52]. The subject of this review is particles reported up to 2024 composed of various organic or inorganic compounds bearing cytC.

## 2. Nanoparticle Carriers of Exogenous Cytochrome *c*

### 2.1. Anticancer Cytochrome-Bearing Particles: Chronology

Cytochrome *c* (cytC) bearing hybrid (nano)particles composed of various substances, which can be absorbed by tumor cells by two main mechanisms, phagocytosis and receptor-mediated endocytosis, reported in the literature by the end of 2024, are given in Table 1 and Table 2.

### 2.2. Mesoporous Silica Nanoparticles (NPs)

Mesoporous NPs have a large specific surface area (hundreds of m^2^/g) and are therefore considered promising carriers for anticancer chemotherapeutics adsorbed in the pores. However, even small globular proteins, such as cytC, cannot penetrate ordinary particles whose pore diameter is smaller than 3 nm. To overcome this problem, Guo et al. (N26 in Table 1 and Table 2) [82] synthesized large mesoporous silica (SiO_2_) particles with a size of about 200 nm, a specific surface area of 260 m^2^/g, and a pore diameter of 10–30 nm, allowing the 3 nanometer-sized cytC globules to penetrate.

The surface of mesoporous silica particles with an electrokinetic potential ζ = −9 mV (determined from the surface silanol groups) was modified by silane coupling molecules which have a pH-dependent amide bond and carboxyl group. The last increases the negative surface potential to ζ = −15 mV, enhancing the electrostatic adsorption of the positively charged cytC globules onto the particle surface. Under acidic conditions, the intramolecular amide bonds split, and protonated amino groups (−NH_3_^+^) emerge, reversing the surface potential from negative to slightly positive: ζ = +1 mV for 30 min and +7 mV after 150 min at pH 6.8. This charge changes from negative to positive conditions through the electrostatic desorption of cytC globules into the endosomes of cancer cells (formed at particle uptake by phagocytosis), where the medium is acidified to pH 5.

The second problem identified in previous studies was the leakage of adsorbed molecules during transport to target cells. To prevent the leakage, the pore entrances were plugged with gold nanoparticles 20 nm in diameter. Their initial negative potential (ζ = −20 mV) was reversed to positive (ζ = +20 mV) by grafting a modified 5 kDa polyethylene glycol (PEG), whose polymer chain ends with an amino group and binds covalently to the gold surface by SH group on the other end. The modified Au-PEG-NH_3_^+^ nanoparticles with a hydrodynamic size of 46 nm were electrostatically adsorbed onto the negative surface of the modified mesoporous silica colloid particles. This plugged the pores after cytC adsorption, forming Au-cytC-SiO_2_ hybrid particles. The gold nanoparticles were subsequently desorbed when the surface charge of the modified silica particles became positive in the endosomes due to the acid-induced cleavage of the amide bond in the silane coupling molecules.

The adsorption and desorption of cytC globules and gold nanoparticles were proven by adsorption spectroscopy and electron microscopy of the Au-SiO_2_ hybrid particles, and the pH-induced surface charge shift was demonstrated by measuring the electrokinetic potential of the particles. The composite Au-SiO_2_ colloid particles achieved encapsulation efficiency and drug loading of 27% and 10%, respectively, compared to 15% and 6% without Au nanoparticles. Within 1 h, the desorption of cytC from Au-cytC-SiO_2_ particles reached 20% at pH 7.4 and 45% at pH 5.0 but was practically absent in the alkaline medium at pH 9.1. After desorption, cytC remained active. In vitro HeLa cell experiments confirmed Au-cytC-SiO_2_ particle uptake via flow cytometry and fluorescence microscopy. The cytotoxicity over 72 h reached 80% at a concentration of Au-cytC-SiO_2_ particles corresponding to 50 μg/L cytC, while the control probe with Au-SiO_2_ particles (without cytC) had a maximal cytotoxicity of 20%. In vivo experiments with mice infected with HeLa cells showed that when the Au-cytC-SiO_2_ hybrid particles were injected into them, the tumor grew more slowly, and its weight after 18 days was one-fourth less than that in the particle-free control group.

This work is not only comprehensive (physicochemical, in vitro, and in vivo experiments) but also interesting due to the use of pH-induced desorption, driven by charge changes resulting from a chemical reaction: The appearance of amino groups after amide bond cleavage. However, one limitation is that the amino bonds split even at neutral pH and the potential decreases to ζ = −3 mV for 30 min at pH 7.4. This conditions the release of cytC globules into the extracellular medium before the Au-cytC-SiO_2_ particles uptake. The pH dependence of the charge change rate and degree, as demonstrated in physicochemical experiments, should provide a faster and stronger desorption of Au nanoparticles and cytC globules into the endosomes of cancer cells after the phagocytosis of Au-cytC-SiO_2_ hybrid particles but a slower and weaker into the blood plasma. However, the plugging effect of Au-PEG-NH_3_^+^ nanoparticles remains unproved because of the absence of a physicochemical experiment comparing the cytC release from Au-cytC-SiO_2_ and cytC-SiO_2_ hybrid particles and an in vitro experiment of their cytotoxicity in cancer cell culture. Although the achieved cytotoxicity (80%) is higher than that reported by most other authors, it remains relatively modest considering the well-designed dual effect: (a) pH-conditioned chemical change of the surface charge of the modified SiO_2_ particles from negative to positive, which enables the intracellular electrostatic desorption of the positively charged cytC globules, and (b) prevention of leakage by plugging the pore entrance with larger Au-PEG-NH_3_^+^ nanoparticles.

It can be concluded that there are two reasons for the limited efficiency. The first is the low drug loading (only 10%) despite the large specific surface area and wide pores. This can be explained by the adsorption of the protein globules at the pore entrance, but their penetration into the depth is hindered due to the electrostatic repulsion from the already adsorbed globules, i.e., cytC adsorption recharges the pore entrance from a negative to a positive potential, preventing free globules from reaching unoccupied areas in the depth of the pores. The second reason is the small charge change: ζ = −3 mV at pH 7.4 and +1 mV at pH 6.8, reached in 30 min and the latter in 150 min: ζ = −2 mV and ζ = +7 mV, correspondingly. These small differences in surface electric potential allowed the desorption of 25% of the cytC globules in the extracellular culture medium, preventing their entry into cancer cells, since the cytoplasmic membrane is impermeable to proteins and cytC can only be introduced via phagocytosis of the Au-cytC-SiO_2_ hybrid particles. The low positive potential of the recharged particle surface is insufficient for complete intracellular desorption owing to the non-electrostatic van der Waals attraction. Even at pH 5.0, desorption reaches only 47% in 2 h at 37 °C and does not increase further with time.

### 2.3. Montmorillonite Nanoplates

CytC-MM hybrid monoplates composed of alumosilicate mineral montmorillonite (MM) and adsorbed cytC (N22 in Table 1 and Table 2) [75,76,77,78] were designed to achieve an anticancer effect without conventional chemotherapeutics. Selectivity is ensured by designing submicron-sized cytC-MM particles that can be phagocytosed by cancer cells and some immune cells (neutrophils and macrophages), whereas normal (healthy) cells lose this ability during differentiation. These particles could be applied directly to skin cancer, administered rectally for colon cancer, or taken orally for other digestive system cancers due to the high resistance of cytC to acid, alkaline, and enzymatic destruction in the gastrointestinal tract. Despite their apparent simplicity, these cytC-MM particles showed 97% cytotoxicity in an in vitro colon cancer cell culture experiment after 96 h of treatment, a value that is one of the highest reported in the literature (Table 2) [75,76,77,78,86]. This result was achieved starting from the physicochemical properties of cytC and carrier particles investigated in [87,88,89,90].

For the easy release of cytC after particle absorption, the cytC globules were adsorbed electrostatically onto MM monoplates, a mineral used in human medicine. Initially, the pH dependence of the net charge of cytC in its native 3D conformation was investigated computationally, which showed that the cytC globules were positively charged in the pH range below pH 10, and then their isoelectric point pI 9.3 was determined experimentally by investigating the pH dependence of the electrophoretic mobility and electric polarizability of the cytC adsorption monolayer on MM colloid plates. Starting from the positive charge of cytC at physiological pH values, MM monoplates were selected as a carrier because they have a negative pH-independent electrical charge, which provides electrostatic adsorption of cytC over the entire range pH 3–9, where cytC globules retain their native 3D conformation both in the free state and adsorbed on different surfaces. The 1 nm thickness of the MM monoplates provides a high protein/mineral ratio for bilateral adsorption: cytC-MM composite particles have a thickness of 7 nm (6 nm of which are due to the cytC globules, each 3 nm thick). For efficient uptake of cytC-MM particles by phagocytosis, 0.4 μm sized MM monoplates were isolated from the initial MM suspension by centrifugation. The size was determined electro-optically from the rotational diffusion coefficient, measured by the rate of disorientation after switching off the applied electric field. The 1 nm thickness of the MM monoplates allows investigation of their mass increment by measuring the static light scattering intensity. The study of the cytC adsorption on MM monoplates by microelectrophoresis and static and electrical light scattering methods showed that under saturated adsorption, the cytC globules form a monolayer with a density of about 8 globules per 100 nm^2^, and 23,700 cytC macromolecules were adsorbed on one MM nanoplate with the size of 0.43 nm. Aggregation of the particles was observed only at a cytC concentration at which the total electrical charge of the cytC-MM particles was close to zero (recharging point of MM plates from negative to positive charge), which appears at 5:3 mg/mg cytC/MM ratio. Above this protein concentration, the total charge was positive, which ensured electrostatic adsorption of cytC-MM plates onto the negatively charged cytoplasmic membrane of cancer cells as a first step in their uptake by phagocytosis. By selecting an appropriate 3 mg concentration of the MM suspension and a 10:3 mg/mg cytC/MM ratio, conditions were found to avoid aggregation, keeping the cytC-MM particle size optimal for uptake by phagocytosis.

### 2.4. Gold-ssDNA Nanoparticles

In the article by Park et al. (N20 in Table 1 and Table 2) [73], the idea is to create gold nanoparticles whose light absorption differs inside and outside cancer cells so that selectivity is ensured by photothermal shock triggered by irradiation with a powerful laser, which kills cancer cells without affecting healthy ones. This effect is based on the light absorption by gold nanoparticles and its conversion into heat energy. The different absorption is possible because the absorption spectrum of gold nanoparticles depends on their size due to the free movement of electrons in the metal, so they absorb strongly at a certain wavelength where resonance (plasmon effect) occurs (unlike dielectric particles where light absorption occurs due to chromophore molecules). Therefore, the authors created hybrid cytC-ssDNA-Au nanoparticles by adsorbing ssDNA and cytC onto spherical gold (Au) nanoparticles. The electrical charge of these composite nanoparticles is pH-dependent: The electrokinetic potential is negative (ζ ≈ −30 mV) at pH 7.4, but nearly zero (ζ ≈ +3 mV) at pI 5.5. Thus, at pH 7.4, the negatively charged Au nanoparticles remain single due to electrostatic repulsion between them, retaining their original size of 100 nm, but aggregate in the endosomes where reversible aggregation occurs due to the acidic medium; then, the peak of the absorption spectrum shifts from 520 nm to 550 nm. This provides partial selectivity for irradiation at 633 nm (He-Ne laser) and complete selectivity at wavelengths λ ≥ 650 nm, where single nanoparticles do not absorb light, but the aggregate absorption is strong due to their large size. The hydrodynamic size, pH-dependent surface potential, and light absorption spectra were demonstrated by dynamic light scattering and spectroscopy in physicochemical experiments with aqueous suspensions of cytC-ssDNA-Au particles. The selectivity was confirmed by in vitro biological experiments: in mouse melanoma (skin cancer) cell cultures, cytotoxicity after irradiation with a powerful diode laser (660 nm, 6 W optical power) was about 75%, whereas cytotoxicity in healthy cell cultures was virtually absent after a 12 h incubation.

This study was well designed as the hybrid gold nanoparticles remain single in the extracellular environment due to their negative surface potential but aggregate in the endosomes (formed after particle uptake by phagocytosis) due to the disappearance of the potential caused by pH lowering as a result of H^+^-pump activity. In addition, the cytoplasm of the cancer cells is slightly acidic due to the lactic acid accumulation because the oxidative phosphorylation in mitochondria does not function. The physicochemical results and the selectivity under photothermal shock are convincing, but the interpretation is scant or lacking. Firstly, the observed pH dependence is not due to the adsorbed macromolecules, since the negative charge of DNA (determined by its phosphate groups) and the positive net charge of cytC (the isoelectric point is pI 9.3 [88]) are practically unaltered in the range pH 5.5–7.4. The pH-induced reduction of the electrokinetic potential is due to the oxide groups on the surface of Au particles emerging at their synthesis in oxygen-containing media. Secondly, not single cytC-ssDNA-Au nanoparticles are absorbed by phagocytosis (because of their too small size of about 10 nm) but their aggregates. The nanoparticle aggregation occurs due to the high ionic strength in the extracellular medium (where the NaCl concentration is 0.15 M), at which the moderate surface electric potential (ζ ≈ −30 mV at pH 7.4) cannot prevent particle aggregation. The major weakness of this work is that cytC was used only to compensate for the negative charge of DNA but not as an initiator of apoptosis. The absence of apoptosis was proved by the lack of cytotoxic effect after 12 h incubation with cytC-ssDNA-Au nanoparticles but without photothermal shock. The reason for the apoptosis absence probably lies in the inability of the positively charged cytC globules to desorb from the DNA adsorption layer, which has a high negative charge density due to the phosphate groups and the flexibility of the ssDNA (single) chains (in contrast to the semirigid dsDNA chains).

### 2.5. Hybrid Iron Oxide–Gold Nanoparticles

Fe_3_O_4_ iron oxide (magnetite) particles have recently been widely used as carriers of various chemotherapeutics because of their ferromagnetic properties: they acquire a magnetic moment in an external magnetic field, unlike Fe_2_O_3_ particles, which are superparamagnetic. As mentioned in Section 2.4, gold nanoparticles possess special optical properties, such as plasmon resonance. The composition of hybrid iron oxide–gold nanoparticles combines the specific properties of both compounds. In cytC-bearing hybrid particles, gold is used as an outer shell that prevents enzymatic degradation of the protein globules and oxidation of the iron oxide core [91].

In two consecutive articles, Hoskins et al. (N7 in Table 1 and Table 2) investigated the combined effect of doxorubicin [59] and other [70] anticancer chemotherapeutics with cytC carried by Au-Fe hybrid nanoparticles composed of a 30 nm magnetic Fe_3_O_4_ core decorated with an Au shell [60,86,92]. The cytC globules are conjugated irreversibly via covalent bonds (SH–Au) of their thiol groups to the gold surface forming cytC-Au-Fe_3_O_4_ hybrid nanoparticles. In vitro experiments with cancer cell cultures showed that the cytotoxic effect of the chemotherapeutics occurs more rapidly when they are adsorbed on Au-Fe nanoparticles (in comparison with their molecular solutions) and further increases when cytC is bound to the particles, i.e., the results disclose that the enhancement of the cytotoxic effect of a given chemotherapeutics is observed due to the involvement of a second cytotoxic effect—apoptosis, whose activation has been demonstrated by various methods. The authors concluded that the cytC-induced enhancement of the cytotoxic effect means a synergism between apoptosis and the anticancer effect of chemotherapeutics. Cytotoxicity varied among the five types of chemotherapeutics and the three types of hepatocellular carcinoma (HCC) cell lines tested, but one cancer line was resistant to both particles carrying only chemotherapeutics and those complexed with cytC. Since the authors assume that cytC binds irreversibly via covalent bonds to the surface of Au-Fe_3_O_4_ nanoparticles, they suggest that apoptosis is activated by some other, unknown up to now, mechanism instead of the previously known one.

It can be inferred that the authors’ assertion of synergism between the cytotoxic effects of the chemotherapeutics and apoptosis is unfounded because cytotoxicity induced by cytC alone (bound to the particles but without chemotherapeutics) was not studied. Synergism means that the total cytotoxic effect exceeds the sum of the effects of cytC and chemotherapeutics separately, i.e., a super-additivity should emerge. The claim of a novel mechanism of apoptosis activation is also unsupported as there is no evidence that cytC globules cannot desorb: These authors provide no proof of cytC-Au covalent bonds. Such bonds can be formed with big proteins such as albumin [93], whose conformational lability provides access to cysteine SH groups on the gold surface. However, cytC has no superficially located SH groups, and its very stable 3D structure prevents conformational change at the Au surface. CytC has only two SH groups, both engaged in covalent bounds with the heme cycle. Due to the stability of these bonds, cytC retains its redox capability by being immobilized both electrostatically and covalently via a carbodiimide linkage to the carboxylic acid-terminated alkanethiol monolayer deposited on gold electrodes [94].

A simpler explanation for the cytC-induced enhancement of the cytotoxic effect of chemotherapeutics can be proposed, instead of the authors’ alternative apoptosis activation mechanism. More likely, cytC globules bind noncovalently to Fe_3_O_4_ particles via electrostatic adsorption, and they are released by desorption into cancer cells, triggering apoptosis by the known mechanism of cytC-Apaf-1 apoptosome formation (Figure 1). CytC desorption is conditioned by the high ionic strength (0.15 M) and increased endosomal acidity (pH < 6). The electrostatic adsorption of cytC globules onto Au-Fe_3_O_4_ hybrid nanoparticles is determined by the positive charge of cytC globules at pH < 9 [88,89] and the negative charge of Au-Fe hybrid particles (ζ = −23 mV) [91]. This type of physical adsorption of cytC is also observed on negatively charged dielectric surfaces [91] and on gold nanoparticles [95], where the electrostatic adsorption is stronger due to the mirror effect of free electrons in the metal. The negative charge on the gold surface arises from the binding of the oxygen atoms of CO_3_^2−^ and NH_2_O^−^, originating from the solutions of Na_2_CO_3_ and NH_2_OH·HCl, used in the synthesis of the Au seeds and the Au shell on the Fe_3_O_4_ core, respectively [91], to the gold surface via coordinate bonds with the Au atoms [96,97]. In addition, the Fe_3_O_4_ core may also contribute to the negative charge at pH 7.4 (the isoelectric point of the Fe_3_O_4_ nanoparticles is around pH 5) [98]. The last possibility is due to the non-dense gold layer on the surface of the Au-Fe_3_O_4_ nanoparticles, which appears to consist of separate domains formed by the chemical reduction of AuCl_4_^−^ anions from the tetrachloroauric acid (HAuCl_4_) solution on 2 nm Au seeds adsorbed on the positively charged groups of the poly(ethylenimine) polymer covering the Fe core. The non-density of the Au layer is indicated by the fact that the adsorption maximum is around 540 nm [91], rather than being shifted into the infrared region 800–1200 nm where the surface plasmon resonance occurs.

Regarding the supposed covalent binding of cytC globules to the Au surface based on a previous investigation [93], it can be noted that in this case, the covalent binding does not occur spontaneously between the SH group of cysteine and the Au surface but by a chemically activated reaction between the NH_3_^+^ groups of lysine residues to the COO^−^ groups of the alkanethiol monolayer. As for the found resistance of one of the cancer lines to Au-Fe_3_O_4_ and cytC-Au-Fe_3_O_4_ nanoparticles (both bearing anticancer chemotherapeutics), it is likely due to the use of a two times lower concentration of nanoparticles and/or the reduced ability of the cancer cells to phagocytose.

### 2.6. Polymeric Nanoparticles

Polymeric nanoparticles are composed of different polymers, including polyethylene glycol (PEG), polypropylene glycol (PPG), triphenylphosphonium (TPP), hyaluronic acid (HA), chitosan, hyperbranched polyhydroxyl (HBPH), etc.

In article [81] (N25 in Table 1 and Table 2), Chinese authors described polymeric nanoparticles as carriers of the chemotherapeutic paclitaxel, which has plant origins and activates the release of cytC from the mitochondria [99]. Although these nanoparticles do not carry cytC, they are included in the review due to their sophisticated structure and high cytotoxicity achieved. Since paclitaxel is poorly soluble in water, the commercial drug Taxol was previously used as an adjuvant; however, it causes a severe hypersensitivity reaction. More importantly, these drugs were poorly effective in cancers with inherent or acquired multidrug resistance. To increase the paclitaxel effectiveness, the authors prepared hybrid polymer particles with an average size of about 140 nm, composed of PEG-PPG-PEG triblock copolymer that forms the particle core, and triphenylphosphonium and hyaluronic acid attached to both ends of the triblock chains via covalent bonds. TPP and HA form a non-dense corona around the core of the polymeric particles, as seen in transmission electron microscopy images. The phosphorus atom of TPP imparts a positive charge to the tetrablock polymer chain TPP-PEG-PPG-PEG, which determines a positive electrokinetic potential ζ = +13 mV, and the carboxyl groups of HA imparts a negative charge (ζ = −25 mV) to the pentablock (TPP-PEG-PPG-PEG-HA) polymeric particles. HA on the outer particle surface binds to the CD44 receptors, which are expressed on the cytoplasmic membrane of many cancer cells [26], leading to receptor-mediated endocytosis. Afterward, inside lysosomes, HA is degraded by hyaluronidase, demonstrated by a charge shift from ζ = −25 mV to ζ = +7 mV in 2 h at pH 5.6. The corona-bare TPP-PEG-PPG-PEG tetrablock particles (released from lysosomes) are incorporated into the mitochondrial membranes, which are hyperpolarized in cancer cells with multidrug resistance. The incorporation of paclitaxel-bearing polymeric particles leads to the release of the mitochondrial cytochrome *c* (cytC) into the cytoplasm, where it forms complexes with Apaf-1, which activate the enzymes caspase-9 and caspase-3. These beginning steps of the intrinsic pathway of apoptosis were observed by confocal laser scanning microscopy with particles, liposomes, and mitochondria labeled with fluorescent dyes.

In vitro experiments with human lung adenocarcinoma and drug-resistant mouse breast cancer cells showed that the commercial drug Taxol had weak activity in multidrug-resistant cases, while HA-coated polymeric particles induced high mortality in both types of cancer cells. In vivo experiments in breast cancer-bearing mice showed increased localization and anticancer activity of the composite particles, more pronounced in HA polymeric particles compared to bare polymeric particles. Since the active anticancer substance paclitaxel is released from the particles depending on the pH of the medium (in 10 h about 25% at pH 7.4 and 60% at pH 4.5), the release is mainly in the liposomes of cancer cells (where the medium is acidic) and probably weak in the blood plasma and intercellular space. For ”normal” cancer cells, HA polymeric paclitaxel-transporting particles showed no advantage over Taxol, as 50% cytotoxicity was reached at the same concentration. However, in multidrug-resistant cancer cells, HA polymeric particles were significantly more effective: 50% cytotoxicity was reached at 10 μM in 48 h, whereas Taxol required ten times higher concentration. At a 50 μM concentration, the cytotoxicity of HA polymeric particles reaches about 90% in 48 h and 95% in 72 h, while that of Taxol remains 50% in the same time ranges of treatment.

This study is impressive in terms of the range and number of methods used, but some aspects of the interpretation of the results remain unclear. The picture looks as follows. Paclitaxel is embedded in the core of the particles, which is hydrophobic because PPG segments of the PEG-PPG-PEG block copolymer are localized there [99]. The hydrophilic PEG segments make up the middle part of the nanoparticles, which is compact due to the high flexibility of the PEG chains. The HA segments make up the outer particle shell of the hybrid particles, which is untight due to the lower flexibility of the HA chains, composed of hexagonal (hexose) nuclei. The dissociated carboxyl groups of HA give the particles a negative charge, preventing adsorption onto healthy cells (whose cytoplasmic membrane is also negatively charged), such as the blood vessel epithelium. Consequently, HA particles will only enter cells capable of receptor-mediated endocytosis, i.e., cells with exposed CD44 receptors, which specifically bind HA. So, the selectivity of the cytotoxic effect of HA polymeric nanoparticles is determined by the ratio of CD44 receptors in cancer and normal cells. In vivo experiments showed that in addition to cancer tissue, the particles also entered other tissues of the mice, most notably the liver.

The authors’ assumption that CD44 receptors mediate particle binding is based on the known ability of this membrane-integrated globular protein to bind HA. Indeed, the fluorescence intensity of composite HA polymeric nanoparticles labeled with a hydrophobic green fluorescent dye was 5-fold higher than that of bare polymer particles (Figure 4a in Ref. [81]) and when the extracellular medium contained a saturating concentration of HA that prevented the binding of HA particles to CD44 receptors due to competitive inhibition. However, the results in Figure 4c in Ref. [81] contradict this inference: The fluorescence intensity and kinetics of both “normal” cancer cells and drug-resistant ones were the same for both types of particles, regardless of the presence or absence of HA on their surface. This suggests that the particles enter the cells in a different way. Furthermore, the CD44 receptor is a specific marker for stem cells (normal and cancer), but the authors have no evidence that CD44 is expressed in their cells. The eventual binding of HA polymeric nanoparticles to CD44 receptors means that in the macroorganism the selectivity will be severely limited since they will not only attack cancer stem cells, which are a negligible percentage (cancer tissue develops from one or two degenerate cells), but will also attack healthy stem cells with all the consequences for the immune and hematopoietic systems as synthetic anticancer chemotherapeutics do.

There are a few more uncertain points in the authors’ interpretation. Firstly, the confocal laser scanning microscopy images are of low resolution: Although many cells with about 20 μm size are visible, their organelles are indistinct. This, together with the inevitable dye diffusion, makes it impossible to conclude with certainty where the polymer particles are located. Because the organelles are stained with different fluorescent dyes, lysosomes (red), nucleus (blue), and mitochondria (red), the authors interpret the yellow staining of the cells as the presence of green fluorescent composite (HA and naked) particles first in the lysosomes and then in the mitochondria, but this interpretation is based rather on the imagination of the authors. It is more likely that the hydrophobic green fluorescent molecules diffuse from the particles into the hydrophobic bilayer of all lipid membranes, most of which are those of the endoplasmic reticulum, so the cells acquire a yellow color regardless of the localization of the absorbed particles.

Secondly, although the authors conclude that cell death occurs via apoptosis, it is possible that this is a concomitant phenomenon to the action of the anticancer substance paclitaxel. No definite conclusion can be drawn since no experiment was performed with the same particles but without paclitaxel. Therefore, the claim that the cytotoxic action of polymeric nanoparticles is due to activation of the intrinsic apoptosis pathway remains an unproven hypothesis. Indeed, the presence of cytC in the cytoplasm and the activation of caspase as the initiation of the apoptosis process have been demonstrated, but this may not be the main cytotoxic effect. It is probable that the effect of the HA polymeric nanoparticles is due to pore formation in the cytoplasmic membrane of the cells resulting from the incorporation of the hydrophobic PPO segments of the tetrablock copolymer TPP-PEG-PPG-PEG into the hydrophobic bilayer of lipid membranes. This process starts with the electrostatic attraction of the positively charged TPP heads of the polymer chains to the negatively charged cytoplasmic membranes after enzymatic degradation of HA of the pentablock copolymer TPP-PEG-PPG-PEG-HA: “stripping” of the particles, in which their electrical charge changes from negative to positive. The incorporation of the naked polymeric nanoparticles causes increased membrane permeability, and this leads to multiple effects. In particular, the hyperpolarization of the internal mitochondrial membrane is abolished and cytC globules escape through their external membrane into the cytoplasm, triggering the intrinsic apoptotic pathway. An indication of the decrease in transmembrane potential is the transition from red to green fluorescence of the mitochondria stained with the used lipophilic cationic dye.

### 2.7. Cytochrome c Particles

In contrast to the previously discussed hybrid particles, Griebenow et al. studied nanoparticles composed of cytochrome *c* (cytC) globules as carriers for anticancer drugs in the hope that the proapoptotic protein cytC would have a synergistic effect by initiating the apoptosis in addition to the action of chemotherapeutics. These investigations continue this research group’s studies on the use of proteins as transporters of anticancer agents [100,101].

#### 2.7.1. Oleic Acid Bearing cytC Nanoparticles

In a study [66], oleic acid (OA), which has cytotoxic effects unrelated to apoptosis (caspase-independent cell death), was used as the active substance in cytC nanoparticles (N13 in Table 1 and Table 2). Previously, these authors demonstrated [102], that particles formed by the milk protein α-lactalbumin and monounsaturated OA do not induce apoptosis. Instead, OA itself has a cytotoxic effect, contradicting earlier studies by Svanborg and co-workers showing that this lipid–protein complex causes apoptosis exclusively in tumor cells. The cytC-OA composite particles were formed at pH 8 (at which pH the OA head is negatively charged due to the complete dissociation of the carboxyl group), a temperature of 45 °C, and an enormous molar ratio of the two components: 67 thousand OA molecules per 1 protein macromolecule [66]. BSA-OA nanoparticles constructed from complexes of OA with bovine serum albumin (BSA) were used for comparison; this protein has no activity but is only a carrier of OA. The particle size of cytC-OA and BSA-OA is about 120 nm and 170 nm, respectively; this size falls in the 100–800 nm range, which is suitable for penetration into cancer cells [103,104]. One cytC-OA particle carries 12 OA molecules and BSA-OA –53 molecules of OA. The authors explained this OA/protein molar ratio by the greater mass of BSA compared with that of cytC, but the zero electrokinetic (zeta) potential of cytC-OA particles indicates that adsorption of OA molecules onto cytC globules is entirely electrostatic (due to the attraction of negatively charged OA molecules to positively lysine residues on cytC); the adsorption stops when the isoelectric point is reached. The significant negative net charge (ζ = −81 mV) of BSA-OA particles indicates that OA adsorption in this case is predominantly hydrophobic (mainly contributed by the fatty chains of OA); this explanation is in agreement with what is known from the literature that the human SA is a major blood protein that is a natural transporter of fatty acids [105]. Considering the ability of cytC to activate the enzymes caspase-3 and caspase-9 in cell lysate, the formation of complexes with OA reduces the stability of the 3D structure of cytC globules. This destabilizing effect increases with temperature: cytC-OA particles formed at 65 °C induced a twofold lower activation of both caspases compared to those formed at 25 °C.

In vitro experiments to measure LC_50_ showed that 50% cytotoxicity was reached at a lower concentration of cytC-OA particles compared to OA, but the difference was not large, indicating that the main effect was of OA molecules, both in the free state or noncovalently bound to a protein macromolecule. At a concentration twice the LC_50_, over 6 h of treatment, cytC-OA particles had a several-fold stronger cytotoxic effect on cancer cell cultures compared to BSA-OA particles, which is comparable to that of free OA. The stronger cytotoxic effect of cytC-OA particles is due to the activation of the apoptosis process, an indicator of the early stages where the activity of caspase-3 and caspase-9 is found to increase by about 50% (their primordial activity in cell lysate was assumed to be 100%). Such an increase in enzyme activity was not present upon treatment with BSA-OA particles, BSA, and OA, and the late morphological changes of dying cancer cells were different in the two types of composite particles. Despite the high efficiency in affecting cancer cells with cytC-OA particles, the problem is that they also have too strong a toxic effect on normal (non-cancerous) cells: cytC-OA particles have a cytotoxicity of about 70–80% at twice the LD_50_ concentration in Cho-K1 cell culture. This cytotoxicity is approximately the same for both types of particles, confirming the conclusion that the active substance is OA. The results indicate that cytC-OA particles do not have sufficient specificity of action, and this makes them unpromising for the treatment of neoplasms because OA–carrier particles will kill all living cells, unlike chemotherapeutics, which kill only dividing cells.

#### 2.7.2. Hyaluronic Acid Bearing cytC Nanoparticles

Selective cytotoxic effects on some neoplasms can be achieved by treating them with cytC particles bearing a ligand for a receptor, which is overexpressed on the external surface of the cytoplasmic membrane of cancer cells but is absent or has a low surface concentration in normal cells. One such receptor is CD44 (also called P-glycoprotein-1), which is a transmembrane glycoprotein with a molecular mass of 85–200 kDa. CD44 has an extracellular domain (ectodomain), a transmembrane domain, and an intracellular (cytoplasmic) domain [106]. The extracellular domain interacts with hyaluronic acid (HA, hyaluronate, hyaluronan), which is its principal ligand. HA is a high-molecular-weight heteropolysaccharide whose repeating basic unit is a disaccharide of D-tolucuronic acid and N-acyl-D-glucosamine; the carboxyl group of the first provides the negative charge of the polymer at physiological pH values.

To induce apoptosis in CD44-overexpressing cancer cells using extracellular cytC, Griebenow and co-authors prepared cytC-HA protein–polymer complexes from positively charged cytC globules and negatively charged HA chains [107]. In a subsequent work [69] (N17 in Table 1 and Table 2), the composition of the particles was improved by crosslinking the cytC globules with dithiobis(succinimidyl) propionate (DSP), which forms disulfide bridges. This prevented the premature cytC release in the extracellular medium, but, after the uptake of cytC-HA-DSP nanoparticles by receptor-mediated endocytosis, the protein globules are released due to the disruption of intermolecular disulfide bonds under the influence of glutathione (GSH), which possesses reducing properties. The GSH concentration differs strongly between the cytoplasm and extracellular medium by four orders of magnitude. Under conditions mimicking the extracellular (1 μM) concentration of GSH, about 20% of cytC macromolecules are released in the first 8 h, and 30% in 24 h at 37 °C. At the same temperature and times, but at 10 mM GSH (imitating its intracellular concentration), the release was about 90% and 100%, respectively.

The formation of cytC-HA-DSP protein–polysaccharide particles was carried out by nanoprecipitation induced by the gradual addition of ethyl alcohol to the solution of cytC and HA at a 1:4 cytC/ethanol ratio, followed by the addition of DSP, freeze-drying, and subsequent resuspension. Circular dichroism spectra showed that this procedure partially altered the tertiary structure of cytC, but it remained physiologically active, activating caspase-9. The activity of this key apoptotic enzyme was about 80% for cytC-HA-DSP nanoparticles and 60% for cytC-HA for protein–polymer complexes relative to that of a molecular solution of cytC (assumed to be 100%) in cell lysate from cancer cells obtained 6 h after addition to the cell culture. At the optimum protein–polymer ratio of 5:2 mg/mg cytC/HA and a concentration of 5 mg/L DSP, the dried particles were spherical with a size of about 0.2 μm in vacuum and 0.8 μm (hydrodynamic diameter) after resuspension in an aqueous solution with 0.15 NaCl at pH 7.4. This size falls in the 100–800 nm range, which is optimal for uptake by endocytosis. Thanks to the carboxyl groups of HA, which predominate considerably over the positive net charge of cytC, the particles are negatively charged: the electrokinetic potential of cytC-HA-DSP particles is ζ = −32 mV, which ensures the stability of the suspension due to the electrostatic repulsion between the nanoparticles. In the optimized preparation procedure, cytC-HA-DSP particles contain about 60% cytC, and up to 40% of the cytC in the suspension is included in the particles. CytC concentration was determined by measuring the optical density of the suspension at 408 nm, which has two components: intrinsic light absorption of cytC and light scattering caused by these submicron particles. To correctly calculate the cytC concentration, the light scattering effect was taken into account by comparison with a suspension of nonabsorbing α-lactalbumine-HA particles, whose optical density spectrum (outside the Soret band) was the same as that of the cytC-HA-DSP suspension.

In vitro experiments with a human lung adenocarcinoma cell line showed that cytC-HA-DSP suspension induced the initial and final stages of apoptosis, respectively, indicated by exposure of phosphatidylserine on the outer surface of the cytoplasmic membrane and by DNA staining of the damaged nucleus. The process begins with the particle penetration into the cancer cells, as shown by a confocal microscope using fluorescent dyes, 6 h after suspension addition to the cancer cell culture. The cytotoxic effect increased eight times with particle concentration (expressed by cytC concentration): from no effect at 0.02 mg/mL cytC to 87% at 0.16 mg/mL cytC (13% viability). The monkey kidney fibroblast control confirms the expectation that a cytotoxic effect is not induced in normal (healthy) cells since cytC-HA-DSP particles do not penetrate them.

The work by Figueroa et al. [69] has a number of advantages: (a) induction of cytotoxic effect via apoptosis alone, without chemotherapeutics; (b) selective effect on cancer cells overexpressing CD44 receptors and no effect on normal CD44-negative cells; (c) predominantly intracellular release of cytC; (d) rapid release of cytC from absorbed cytC-HA-DSP particles within a few h; (e) rapid cytotoxic effect developed in just 6 h; (e) relatively high cytotoxic effect reaching about 90%; (f) use of natural polymer (HA) degradable by the lysosomal enzymes; (g) submicrometric size of protein–polymer particles optimal for receptor-mediated endocytosis.

The results are excellent from the physical chemistry aspect, but the authors’ approach is not very well founded in terms of selectivity. The cytC-HA-DSP particles have only been tested in vitro on selected cancer and normal cell lines with the overexpression or absence of CD-44 receptors, respectively. However, the CD44 receptor is ubiquitously expressed throughout the body: It is widely distributed in normal adult and fetal tissues, where CD44 is involved in adhesive cell–cell and cell–matrix interactions, as well as in cell migration and cell homing [108]. The standard isoform sCD44 was originally isolated from hematopoietic cells but is now found in a variety of tissues: central nervous system, lung, and epidermis, including macrophages, polymorphonuclear leukocytes, erythrocytes, fibroblasts [105]. Normal (non-cancerous) cells, including vital cells of the hematopoietic and immune systems, such as phagocytic cells of the reticuloendothelial system, will die from apoptosis triggered by the extracellular cytC introduced by the receptor-mediated endocytosis if the suspension of cytC-HA-DSP particles is administered in vivo by venous or muscle injection. In particular, it was shown in vitro with human dermal fibroblasts that fluorescently labeled HA, added to the culture medium, was rapidly attached to the cytoplasmic membrane, and, after a minute, it was observed within the cells in association with the CD44 receptor and is transported mostly to lysosomes, as revealed by flow cytometry and confocal microscopy; after 10 min incubation, both HA and CD44 were accumulated in discrete globules in the perinuclear area within the cytoplasm [109].

Thus, this work by Figueroa et al. [69] on the synthesis of HA-cytC-polymer particles, very successful from a physicochemical point of view, does not appear to have significant prospects, as this approach can only provide limited selectivity based on the difference in the expression level of the cellular hyaluronic acid-binding receptor CD44.

#### 2.7.3. Folic Acid Bearing cytC Nanoparticles

Griebenow’s research group has developed a method [58,79,83,110] for creating submicron cytC-FA particles, composed of a core of cytC and a polymer shell bearing covalently bound folic acid (FA) to selectively deliver cytC into cancer cells by binding to alpha folate receptors (α-FR) exposed on the external surface of the cytoplasmic membrane. Here, we review two recent articles (N23 and N27 in Table 1 and Table 2) that are impressive in the range of methods used (physicochemical, in vitro, and in vivo) and the results obtained. Selectivity is determined by the overexpression of α-FR, as shown by about 40% of cancers [111], in which folic acid is involved in enhanced DNA synthesis and modification, in particular in thymine synthesis (one of the four bases of DNA), and in cellular proliferation [112]. α-FR expression is associated with tumor stage and survival, specifically in lung adenocarcinoma [113]. The overexpression of α-FR in tumor cells promotes folic acid ligand–receptor association and subsequent absorption of cytC-FR nanoparticles via receptor-mediated endocytosis [114]. These cytC-FA particles are advantageous over other cytC–carrier particles in that they have a fully cytC-constructed submicron-sized core containing a large number of cytC globules. This allows reaching the critical cytoplasmic cytC concentration threshold with the uptake of a small number of cytC-FA particles into the cancer cell, above which complexes of cytC with the cytoplasmic protein Apaf-1 initiate the caspase cascade [115,116].

In the protein core synthesis, cytC globules bind via a crosslinker, forming covalent bonds with the amino groups of lysine residues of neighboring cytC macromolecules, preventing following cytC-FA particle degradation in an aqueous medium. The crosslinker molecules contain a disulfide bond that can be broken in the presence of a reductant such as glutathione in the cytoplasm, leading to intracellular breakdown of the protein core to individual cytC macromolecules. Since the concentration of glutathione is 10 mM in the cytoplasm, whereas it is only 1 μM in the extracellular medium, this huge difference of 4 orders of magnitude ensures a preferential intracellular release of cytC. Thus, cytC-FA nanoparticles have a threefold advantage over particles with a passive core carrying adsorbed cytC macromolecules: (a) the particle core is composed only of cytC globules, which ensures a high content of this proapoptotic agent in a single particle with submicron size; (b) specific delivery of cytC into cancer cells via receptor-mediated endocytosis; and (c) release of cytC macromolecules only after uptake by cancer cells.

The article by [79] is devoted to the optimization of the technology. The solvent nanoprecipitation method [117], in which an organic solvent is added to an aqueous solution of a protein, was used to create submicron cytC particles. In the prepared aqueous–organic emulsion, the protein globules are concentrated in the water droplets, in which the small intermolecular distances are a prerequisite for protein–protein binding by linear crosslinker molecules. Obviously, this method is only applicable to proteins with a stable 3D structure, preventing protein globule denaturation. When the polypeptide chain unfolds from the 3D globule, it adsorbs onto the interface of the two phases: the hydrophobic amino acid residues (previously located in the globule core) fall into the organic solvent, while the hydrophilic residues remain in the aqueous medium. The stable 3D structure of the cytC macromolecule is ensured by the presence of a heme around which the polypeptide chain is coiled and covalently linked. For this first phase of cytC particle synthesis, the authors found the most suitable organic solvent, its ratio to the aqueous phase, and the optimal cytC concentration in order to obtain cytC particles with submicron size. By testing four types of organic solvents, the authors identified that the addition of acetonitrile to an aqueous solution of cytC with concentrations of 5 μM and 10 μM in a ratio of 1:4 to 1:8 water/acetonitrile was suitable. In the second phase of particle synthesis, the cytC globules in the aqueous droplets were covalently linked to a linear crosslinker, forming multimolecular aggregates that do not dissociate after the removal of the organic solvent. This produces cytC particles of 160 nm or 250 nm size, respectively at concentrations of 5 μM and 10 μM cytC in the initial aqueous solution.

The degradation of the protein core of cytC-FA particles into individual cytC globules under the action of glutathione was confirmed via a physicochemical experiment mimicking the intracellular medium: After 10 h at pH 7.4, cytC release reached about 85% at a concentration of 10 mM glutathione, as in the cytoplasm. Under the same conditions, but at 0.001 mM glutathione (concentration in the extracellular medium), about 15% and 25% of cytC macromolecules are released from the cytC-FA particles with an external size of 250 nm and 350 nm, respectively. The glutathione-induced degradation leads to the release of approximately one-half of the cytC globules in 1 h, and this, together with the large number of cytC macromolecules introduced into the cancer cell upon uptake of a single cytC-FA particle, ensures rapid caspase cascade activation.

During the third phase of hybrid cytC-FA particle synthesis, the protein core is coated with a polymer, increasing the hydrodynamic size of the particles to 250 nm and 350 nm from the initial 160 nm and 250 nm protein cores formed from a 5 mM and 10 mM aqueous cytC solution, respectively. Therefore, a copolymer SH-PLGA-PEG-FA, composed of poly(lactic-co-glycolic acid) (PLGA) and poly(ethylene glycol) (PEG), with a thiol group (SH) and FA are covalently linked to the ends of the chain. Through the SH groups, the copolymer chains bind covalently to the amino groups of the lysine residues. The FA residue at each polymer chain end provides specific binding to α-FR receptors on cancer cells by ligand–receptor interaction. This was demonstrated in experiments with fluorescently labeled PEG-FA chains: both cytC-FA particles and single cytC-PEG-FA macromolecules (two polymer chains covalently linked to one cytC globule) and free PEG-FA chains (without cytC) were absorbed by the cancer cells, whereas PEG chains (without folic acid FA) do not penetrate them. Since the hydrophilic PEG-FA chains and single cytC-FA protein–polymer complexes remain in macromolecular form without forming nanoparticles, this is evidence that the uptake of the SH-PLGA-PEG-FA particles (designated here as cytC-FA) occurs by the mechanism of receptor-mediated endocytosis.

The binding of cytC macromolecules to the PLGA-PEG-FA polymer chains decreases the cytC activity as a proapoptotic agent, as indicated by caspases-3,7,10 activity in cancer cell lysate. Assuming 100% activity when free cytC was added, the activity was about 45% for a single cytC globule with a covalently linked polymer (mono-cytC-PEG-FA, 1:2 protein–polymer ratio) and about 90% for cytC-FA particles (multi-cytC-PLGA-PEG-FA) with a protein core of 160 nm and an outer size of 250 nm. Particles with a 250 nm core and an outer diameter of 350 nm exhibited slightly higher activity. The authors suggest that the cytC core is better protected from polymer chains in smaller particles due to their greater surface curvature. However, the higher activity of the larger cytC-FA particles is more likely due to the smaller percentage of cytC globules, located on the surface of the protein core: the ratio surface/volume decreases linearly with diameter. This happens when cytC-FA nanoparticles are formed in the water/organic emulsion; then, the polymer chains are located at the phase boundary: PLGA segments in the organic phase and PEG-FA segments, together with the cytC associates, in the aqueous phase. This explanation is confirmed by the found twofold decreased activity of the single cytC-polymer complexes in comparison with free cytC globules.

The in vitro cytotoxicity on Levis lung carcinoma (LLC) cell cultures was approximately 50% for mono-cytC-PEG-FA protein–polymer macromolecules (two PEG-FA chains linked to one cytC globule) after 6 h of treatment at a cytC concentration of 120 μg/mL. Free cytC macromolecules (not carrying FA) and PEG-FA polymers (without cytC) had no cytotoxic effect. The cytotoxicity of cytC-FA nanoparticles (multi-cytC-PLGA-PEG-FA complexes) with an external size of 250 nm, tested on LLC and HeLa (cervical cancer) cells, both overexpressing α-FR receptors, reached up to 85% at a maximum concentration of 180 μg/mL cytC after 6 h. Extending the treatment to 12 h did not increase cytotoxicity. At the same concentration (120 μg/mL cytC), cytC-FA particles have a twice stronger cytotoxic effect than the mono-cytC-PEG-FA macromolecules. The fact that single cytC-PEG-FA protein–polymer complexes have cytotoxic effects confirms that cytC-FA particles are absorbed by the mechanism of receptor-mediated endocytosis since uptake by phagocytosis requires particles to be submicron in size. The presence of a protein core in cytC-FA particles leads to higher cytotoxicity; as the critical cytC concentration in the cytoplasm is reached more rapidly, the caspase cascades. The control with mouse embryonic fibroblasts showed no cytotoxic effect, but that with human fibroblasts gave approximately 20% cytotoxicity, an indication of limited cytC-FA particle selectivity.

Since the PLGA-PEG-FA copolymer is amphiphilic, after acetonitrile removal from the aqueous–organic emulsion, the polymer chains in the fully aqueous suspension of submicron-sized cytC-FA particles form nanoparticles with a hydrophobic PLGA core and a hydrophilic PEG-FA periphery. Folate residues remain exposed on the surface, increasing the hydrodynamic size by 100 nm. The formation of polymeric nanoparticles due to the hydrophobic PLGA segments is unnecessary because the polymer chains are covalently linked to the lysine residues of the protein core, and the presence of FA at the end of the hydrophilic PEG chains is sufficient for the absorption of cytC-FA particles by receptor-mediated endocytosis. Therefore, in the following work (N27 in Table 1 and Table 2), Griebenow and co-workers used an FA-PEG polymer without PLGA segments, avoiding nanoparticle formation. The binding of this hydrophilic polymer to cytC then resulted in an insignificant increase of the protein core with 5 nm (hydrodynamic size). These cytC-FA submicron particles synthesized according to the multi-cytC-PEG-FA formula (instead of multi-cytC-PLGA-PEG-FA) have certain advantages over those described above as follows.

Firstly, the release of cytC from multi-cytC-PEG-FA particles with an outer size of 170 nm and a protein core of 165 nm was faster at 10 mM glutathione (corresponding to its cytoplasmic concentration); in only 1/2 h, 75% of cytC macromolecules were released, and 95% was released in 20 h. At a concentration of 0.001 mM glutathione (corresponding to the extracellular medium), the release was slower: negligible in 1/2 h, only about 10% in 20 h, and 20% in 46 h. Secondly, the released cytC reached 95% (compared to the native), as assessed by caspases-3,7 activity in LLC cancer cell lysate, which increased about 2.7-fold upon cytC addition, i.e., no inactivation of cytC macromolecules was observed due to their covalent binding to the crosslinker and PEG-FA polymer. Third, 90% in vitro cytotoxicity was reached after 24 h in cell culture of LLC cancer at the maximal cytC-FA particle concentration, corresponding to 300 μg/mL cytC. Cytotoxicity increased with particle concentration, starting at 75% at 25 μg/mL cytC. At 12 μg/mL, the cytotoxic effect is absent, probably because the cytC-Apaf-1 apoptosome formation threshold was not reached in 24 h. In vivo experiments in mice with lung carcinoma showed that cytC-FA particles penetrated into cancer tissue only 5 min after intravenous injection and effectively suppressed tumor growth.

### 2.8. Ferritin Nanocapsules

Ferritin is an intracellular protein whose subunits form vesicles in which iron is stored, suggesting its use as a carrier for different anticancer agents. The work N19 in Table 1 and Table 2 [72] proposes a novel approach for intracellular delivery of cytC: encapsulation in ferritin capsules, which bind to the CD71 (TfR1) cellular receptors (highly expressed by cancer cells) and are then absorbed via receptor-mediated endocytosis.

In some organisms (archaea, fungi, plants, animals) ferritin consists of 24 subunits, forming a capsule with an outer diameter of 12 nm and an inner diameter of 7 nm, in which iron accumulates as phyrihydrooxide. To use ferritin for transport, its subunits must dissociate and then associate in the presence of drug substances or small proteins, cytC in this case. The problem is that the human H-ferritin dissociates in extremely acidic (pH < 2) or alkaline (pH > 10) mediums in which most proteins denature. Therefore, the authors used ferritin from Archaeglobus figidus, which dissociates at neutral pH and associates in the presence of bivalent ions, forming capsules of 24 subunits with four cavities of 4.5 nm in size that can accommodate the 3-nanometric cytC globules.

The second problem is that the archaea ferritin does not bind to the human cell receptor. To address this, the authors used a recombinant protein (synthesized in *E. coli*), whose polypeptide chain includes the archaebacterial ferritin and a peptide of 12 amino acid residues with which human H-ferritin binds to the human CD71 receptor. The monomers of this ferritin chimera associate at pH 7.4 upon the addition of 50 mM MgCl_2_, forming four cavities of one ferritin capsule that should incorporate four cytC globules. However, even at a 20-fold higher molar concentration of cytC in the ferritin solution, encapsulation was not efficient enough: The cytC/ferritin ratio was only 1.4:1 instead of the expected 4:1. To improve encapsulation, the authors chemically modified a fraction (no more than one-half) of the cysteine residues of chimeric ferritin by S-carboxymethylation so that the attached carboxyl groups added up to 12 additional negative charges, which enhanced the electrostatic binding of the positively charged cytC globules. Due to this improvement, the cytC to ferritin ratio increased 3.2 times, i.e., three cytC globules are included in the four ferritin cavities.

The cytC encapsulation was demonstrated by size-exclusion chromatography (the leakage time of ferritin capsules and free cytC was 5.5 min and 8.3 min, respectively), and detection was demonstrated by the light absorption of ferritin and cytC and the fluorescence of their labeled analogs. The action of chimeric cytC-ferritin capsules was studied in vitro in an acute promyelocytic leukemia cancer culture. The uptake of the protein capsules was demonstrated by microscopy of fluorescently labeled ferritin and cytC, and the occurrence of apoptosis was demonstrated by chromatin condensation and staining with a dye penetrating dead cancer cells.

The main drawback of this approach is the lack of full selectivity since the CD71 receptor is expressed not only in cancer cells but also in all rapidly dividing healthy cells, including those of the hematopoietic and immune systems, and basal cells of mucous membranes and skin [118,119]. No control was made with normal cells in this study. Secondly, the authors’ argument for cytC import via receptor-mediated endocytosis is that this avoids acidification and proteolytic enzyme action in the endosome, as in the phagocytosis of submicron particles. This argument is inapplicable since, in both types of endocytosis (phagocytosis is a type of endocytosis: the uptake of solid particles), the proteolytic enzymes are incorporated into endosomes after their fusion with the lysosomes. The endosome medium is then acidified to pH 4–5 due to the continued action of proton pumps embedded in the endosome membrane, considering that endosomes are formed by folding of the cytoplasmic membrane where H^+^-pumps create the proton gradient. The fact that the absorbed cytC-ferritin capsules successfully triggered the apoptosis is due to the following factors: (a) a very stable 3D structural conformation of cytC in a wide pH 3–12 range [120,121] (its acid denaturation undergoes only at pH ≤ 2 [122,123], which extremely low values are not reached in endosomes) and (b) cytC is stable upon the action of proteolytic enzymes: only 20% of cytC macromolecules are destroyed in a solution of trypsin, chymotrypsin, and elastase in 4 h treatment [124].

### 2.9. Cationic Dextrin

Indian authors (Sarkar et al.) have prepared cationic dextrin nanoparticles as cytC carriers and have reached high cytotoxicity investigated by an impressive set of experimental techniques (N29 in Table 1 and Table 2) [85]. Dextrin was chosen as a biocompatible and biodegradable polymer. Because the dextrin is uncharged, it was chemically modified via the incorporation of quaternary ammonium groups: the derivate is cationic dextrin (CD). To encapsulate cytC by CD, the protein–polymer complex is prepared at pH 11 where the net charge of cytC globules is negative, taking into account its pI 10. Tripolyphosphate is used as an anionic crosslinker. Its negatively charged molecules form electrostatic complexes with the positively charged quaternary ammonium groups of CD. The preparation of the protein–polymer nanoparticles was carried out in a water/acetone mixture in a ratio of 1:1. As a result, spherical cytC-CD nanoparticles with a diameter of 100 ± 30 nm and a zeta potential of +6 mV were formed. The loading efficiency (the percentage of cytC macromolecules included in the nanoparticles) was about 20%; i.e., 80% remained free and lost afterward. The authors explained the use of acetone in order to remove the unencapsulated cytC, but the correct explanation is that the organic liquid deteriorates the physicochemical properties of the medium as a polymer dissolvent. As a result, the dextrin chains are collapsed into nanoparticles.

CytC release from the cytC-CD nanoparticles was found to be linear with the time. At pH 5.5, the release is about 40% within 30 min and reaches almost 90% within 24 h. At pH 7.4, the released protein is 25% and 75%, respectively, within 30 min and 24 h. This pH-dependent difference suggests that cytC release from the cytC-CD nanoparticles will be slightly faster in endosomes where the medium is acidified due to the action of proton pumps. The UV spectra of circular dichroism (minima at 210 and 223 nm, which correspond to Trp-59) show that the cytC released for 24 h at pH 5.5 is structurally unchanged. Its functional activity is validated by the kinetics of the enzyme reaction of oxidation by hydrogen peroxide compared with that of free cytC.

An in vitro experiment with HeLa cancer cultures showed that cells treated separately with cytC solution, chloroquine, and cytC-CD suspension did not exhibit cytotoxic effects within 24 h. However, cytotoxicity emerged with the combined application of cytC-CD nanoparticles and 0.1 M chloroquine, reaching about 45% at a nanoparticle concentration of 1 mg/mL and increasing to 88% at 1.5 mg/mL. In a culture of lung carcinoma epithelial cells A549, the cytotoxicity was about 30% and 60% with treatment with cytC-CD suspension with a concentration of 1 mg/mL or 1.5 mg/mL, respectively. It can be noted that the authors used chloroquine with a concentration of 100 μM, which is in the range 64–128 μM where chloroquine causes apoptosis, as shown in in vitro experiments [125]. The absence of cytotoxicity in the authors’ experiments with separately applied chloroquine or cytC-CD nanoparticles is probably due to not reaching the threshold at which apoptosis is initiated in HeLa cells.

Chloroquine is a well-known antimalarial drug with potential application in cancer and viral therapy due to its penetration into endosomes and subsequent lysis because of increased osmotic pressure [126,127,128]. According to the authors’ explanation, this effect allows cytC to escape from endosomes before their fusion with lysosomes, thus preventing its degradation by proteolytic enzymes. The first part of this explanation looks reasonable because the small chloroquine molecule, being electrically neutral (only 10% of the molecules are positively charged at pH 7.4, considering their isoelectric point pI 8.5), penetrates easily through lipid membranes due to its hydrophobic cycle, but, in the acid medium of the endosome (pH 5.5), its chargeable nitrogen group obtains a positive charge, so the chloroquine molecules become captured in the endosome. However, avoiding fusion with lysosomes as a condition to retain the native conformation of cytC macromolecules is contradicted by all other experiments observed in this review, which described a strong cytotoxic effect caused by apoptosis without using chloroquine or any other agents causing endosome lysis.

The mechanism of the cytotoxic effect induced by a 1.5 mg/mL suspension of cytC-CD nanoparticles in the presence of 0.1 mM chloroquine was investigated employing a great number of techniques, which can disclose the successive stages of the process. Nanoparticle uptake by cancer cells was visualized by fluorescence microscopy of a HeLa culture using FITC-labeled cytC-CD nanoparticles and DAPI; the last dye binds to nucleic acids, attributing them with blue fluorescence. After a 6 h treatment, the green fluorescing nanoparticles were located around the blue nuclei of the cancer cells, indicating their intracellular location.

Further stages of the process are entirely intracellular. To determine the cause of cell death following the combined treatment of HeLa cancer cells with a suspension of 1.5 mg/mL cytC-CD nanoparticles and 0.1 mM chloroquine, key apoptosis indicators were examined: phosphatidylserine translocation to the outer cytoplasmic membrane (early phase), mitochondrial depolarization, nuclear fragmentation, and membrane blebbing (late phase). Apoptosis initiation was confirmed with a Western blot, which detected the cleavage of procasepase-9 into two parts: two distinct protein bands emerged after 6 h treatment with cytC-CD nanoparticles with a concentration of 1.5 mg/mL in the presence of 0.1 M chloroquine.

The early and late phases of apoptosis were investigated by flow cytometry and confocal fluorescence microscopy with a standard fluorescent set: annexin-V and propidium iodide. Annexin-V, labeled with the green fluorescent dye FITC, serves as an early-phase apoptosis indicator due to its specific binding to phosphatidylserine (PS). Biological membranes comprise approximately 20% PS, which is located in the inner cytoplasmic membrane bilayer. During early apoptosis, PS molecules transfer to the outer bilayer, leading to green fluorescence in cells treated with annexin-FITC solution. The late apoptosis phase, characterized by nuclear fragmentation and chromatin condensation, was identified by propidium iodide (PI) staining. PI is a membrane-impermeable dye that penetrates only cells with damaged membranes, binds to nucleic acids, and emits blue fluorescence. Thus, blue nuclei indicate late-stage apoptosis, whereas healthy cells with intact membranes remain unstained. The similar number of cells labeled with annexin-V and PI suggests apoptosis as the primary cytotoxic mechanism rather than necrosis, which lacks early PS asymmetry loss.

Flow cytometry revealed that the cytotoxicity of HeLa cells treated with 1.5 mg/mL cytC-CD nanoparticles and 0.1 mM chloroquine was insignificant after 6 h incubation but strongly increased with time: the percent of viable cells is only 5–7% after 12 h and 18 h incubation. The absence of cells with strong PI fluorescence in combination with the weak annexin-V-FITC fluorescence of the majority of the cells discloses that cell death is caused by apoptosis, not by necrosis. The green/blue fluorescence ratio shows a time-dependent shift in apoptosis phases: After 12 h, 30% of cells were in the early apoptotic phase, but this dropped to 15% after 18 h. As early apoptosis decreased, late apoptosis increased, with 10%, 60%, and 75% of cells in the late phase after 6, 12, and 18 h, respectively.

In the cancer cell, the mitochondrial membrane is hyperpolarized, preventing the leakage of cytC to the cytoplasm in response to initial apoptotic stimuli. The activation of the caspase cascade leads to mitochondrial membrane depolarization—another indication of apoptosis. The transmembrane potential can be indicated by the fluorescence of the cationic lyophilic dye JC-1, which can selectively penetrate the intact mitochondria membrane and change the emitted light wavelength depending on its intramembrane concentration: from green fluorescence in monomeric form to red at high concentration, when the JC-1 molecules form reversible J-aggregates. This allows estimating the mitochondrial potential by the ratio of green/red fluorescence. Flow cytometry experiments showed that about 90% of the fluorescence of untreated HeLa cells was red—an indication of hyperpolarization of the mitochondrial membrane of cancer cells. However, when HeLa cells were treated with 1.5 mg/mL cytC-CD nanoparticles and 0.1 mM chloroquine, the red fluorescence decreased to 80% after 6 h and 50% after 18 h, confirming the time-dependent mitochondrial membrane depolarization.

Confocal fluorescence microscopy of HeLa cultures, colored with JC-1 dye, confirmed the above conclusions. In untreated cells, depending on the light filter used, images appeared red, green, or yellow (their combination), indicating the presence of JC-1 in both aggregate and monomeric forms due to mitochondrial hyperpolarization. However, in HeLa cells treated with cytC-CD nanoparticles and chloroquine, no red fluorescence was observed. The cells appeared entirely green, confirming that JC-1 transitioned from its aggregated form to monomers as a result of mitochondrial membrane depolarization.

The late phase of apoptosis, characterized by chromatin granulation and loss of nuclear membrane integrity, is detected by staining the cell nuclei with the blue fluorescent dye DAPI, which only penetrates cells with damaged cytoplasmic membranes. Confocal microscopic images of HeLa cultures showed structural changes in the nuclei of HeLa cells treated with 1.5 mg/mL cytC-CD nanoparticles and 0.1 mM chloroquine, whereas untreated cells show nuclei with intact morphology. The observed appearance of apoptotic bodies, formed by blebbing of the cytoplasmic membrane, is a unique indication of apoptosis in the late phase. Indeed, SEM images confirmed the emergence of extensive membrane blebbing and a great number of apoptotic bodies on the outer membrane of the treated HeLa cells, whereas no such structures were observed in untreated cells. Therefore, the results obtained by confocal fluorescent microscopy and scanning electron microscopy, convincingly confirmed the appearance of morphological chains in cytoplasmic membranes and cell nuclei, which are specific to the late apoptotic phase.

### 2.10. Particles Bearing cytC: Summary

One of the most commonly used and fashionable particles in the recent past are mesoporous inorganic particles and their various modifications [129,130]. The idea of using them arises from their large internal area (measured by low-temperature nitrogen adsorption), which suggests the intraparticle adsorption of a large number of cytC macromolecules in the pores. Therefore, considering the limited absorption rate of particles by phagocytosis or receptor-mediated endocytosis, this should allow reaching the apoptosis threshold by introducing a smaller number of cytC-bearing nanoparticles into the cancer cell. This presumes an undoubted advantage of mesoporous particles over those with a smooth surface due to the faster formation of cytC-Apaf-1 complexes that initiate the caspase cascade. However, the porous particles did not meet expectations: In vitro experiments with cancer cultures showed no significantly higher cytotoxicity (Table 2, numbers 4, 11). The reason is that during the preparation of the composite particles cytC globules enter only the beginning of the pores, which results in a smaller amount of adsorbed protein because of the pores blockage: geometrically in the narrow pores and electrostatically in the wider ones. Additionally, once inside the cancer cell, the desorption of these cytC globules is hindered (see Section 3.1).

Other types of particles have been used to introduce cytC: aggregates of coagulated proteins [66] and gold-coated hybrid iron particles [59,60,70]. There is a tendency among researchers to synthesize as many sophisticated particles as possible, but this does not result in higher cytotoxicity: Modification of the carriers increases the cytotoxic effect by only 5% (N4, N7, N17 in Table 1 and Table 2). Conversely, 97% cytotoxicity was achieved using smooth flat nanoparticles, carrying only cytC without combination with an anticancer chemotherapeutic (N22) [78].

Various anticancer chemotherapeutics were additionally adsorbed to cytC-carrying particles to enhance cytotoxicity (N9, N17, N18 in Table 2). Undoubtedly, this approach is more effective, achieving up to 80% cytotoxicity, although it remains below the 97% obtained using cytC alone (N22 in Table 2). However, the cytC chemotherapeutic combination decreases the selectivity because, after the destruction of cancer cells, the chemotherapeutic molecules diffuse and can enter the healthy cells surrounding those of neoplastic tissue.

It is important to note that, despite their optimal physicochemical properties, most of the investigated (nano)particles have limited clinical application because very few are approved for application in human medicine. Among the few exceptions are colloidal montmorillonite (bentonite) particles, which have been used for years for various gastrointestinal disorders [131].

## 3. Required Properties for Cytochrome Carriers

### 3.1. Rate of Endocytosis

There is a limit of cytoplasmic cytC concentration below which apoptosis is not triggered because the formation of cytC-Apaf-1 complexes must be faster than their degradation. Thus, the cancer cell must absorb a larger number of particles, each carrying a larger number of cytC macromolecules. Phagocytosis and receptor-mediated endocytosis are energy-dependent processes that require using ATP to bend the cytoplasmic membrane and envelope the cytC–carrier particle adsorbed on it. The ATP concentration in cancer cells is severely reduced because of its rapid consumption in the synthesis of proteins, lipids, and carbohydrates, which is required for fast and uncontrolled cell division, a characteristic of cancer tissue. More importantly, in cancer cells, ATP synthesis is limited to the low-energy process of glycolysis, while the high-energy tricarboxylic acid cycle (Krebs) cycle and oxidative phosphorylation are blocked (Warburg effect).

The cytoplasmic particle concentration is determined by the rates of endocytosis and exocytosis. The ratio between these opposite processes is determined by the geometric and electrical properties of the particles and their extracellular concentration. The dependence on particle size and shape is conditioned by the limited ability of the cytoplasmic membrane to bend to envelope the attached cytC-bearing particle with a nano- or submicron size. The electric charge influences the probability of particles to be adsorbed onto the negatively charged cytoplasmic membrane, considering that they undergo Brownian motion, which competes with the electrostatic attraction when the particles are positively charged and decreases the perimembrane concentration of negatively charged particles. The endocytosis rate is important because the absorbed particles are initially located not in the cytoplasm but in the endosomes, which later fuse with lysosomes. CytC then undergoes the action of proteases (protein-degrading enzymes), which possibly reduce its ability to bind to the cytoplasmic protein Apaf-1, forming active cytC-Apaf-1 complexes, which start the caspase cascade at the first stage of the apoptosis. So, the uptake of a bigger number of cytC-bearing particles per unit of time increases the probability of reaching the apoptosis threshold.

The above consideration leads to the requirement for each carrier particle to adsorb a larger number of cytC globules, but their capability is limited by two factors. Firstly, the density of the protein monolayer (the number of protein globules per unit area) is restricted by both the physical size of the 3-nanometer cytC globules and their random location on the particle surface. Secondly, there is an inability to form a second monolayer because of the electrostatic repulsion of free cytC globules from the first monolayer; the formation of only one cytC monolayer on montmorillonite monolayer was experimentally found [87]. Increasing the surface area by particle size is undesirable because of the difficulty in the endocytosis-mediated uptake of larger particles owing to the need for a large area of cytoplasmic membrane required to envelop the particle. For the same reason, the aggregation of particles should be avoided, although aggregates carry a large number of cytC globules.

### 3.2. Particle Geometry: Surface, Size, and Shape

Using porous particles, even with pores wider than 6 nm (twice the size of cytC globules), does not result in higher cytotoxicity (Section 2.2). The explanation lies in the stronger adsorption and more difficult desorption at the entrance of the pores, both caused by the higher (negative in mesoporous silica) surface electric potential (caused by the overlap of the double electric layers on the opposite pore walls) compared to the outer surface of the particle. This results in adsorption of protein globules only at the pore entrance, where they are electrostatically attached to the negatively charged centers of the pore walls. New cytC globules cannot penetrate deeper into the pore owing to electrostatic repulsion from the positively charged (at pH ≤ 9) cytC globules already adsorbed at the pore entrance. Thus, the effective adsorption area for globular proteins such as cytC is dramatically diminished compared to the total area measured by the low-temperature adsorption of small uncharged nitrogen molecules. Hence, porous particles have a lower capacity for cytC delivery. In contrast, particles with smooth surfaces, such as flat montmorillonite monoplates, allow easy adsorption and desorption, reaching a higher protein layer density due to the lack of additional steric and electrostatic restrictions.

It is assumed that different-sized particles enter the cell by different mechanisms. Particles smaller than 0.2 µm are absorbed by receptor-mediated endocytosis, whereas larger submicrometric particles are absorbed by phagocytosis [132]. The optimal size for uptake by phagocytosis in colorectal cancer cells is between 0.2 and 0.5 µm [133,134]. Particles smaller than 100 nm are also absorbed but rapidly expelled from the cell by exocytosis. Within a few h, 67% of 100 nm particles are ejected from cells by exocytosis, whereas only 37% of 0.6 µm particles leave the cytoplasm of cancer cells [135]. Clearly, for the intracellular introduction of cytC during phagocytosis of one particle, it is desirable to use particles with bigger sizes (to adsorb a bigger number of cytC globules) but limited within the optimal size range.

Particle shape also plays a role in both endocytosis and exocytosis. Asymmetric particles (cylinders) are absorbed faster than cubic particles of the same size [136]. However, the shape dependence cannot be considered separately from particle size and surface nature. Moreover, the type of cancer cells, their size, cytoplasmic membrane composition and elasticity, and the different mechanisms of particle uptake must be taken into consideration when choosing particles for cytC delivery. For this reason, conflicting data exist in the literature. In colon cancer cells, for particles of the same chemical nature and size of 0.2 µm, anisodiametric (disks and cylinders) are absorbed more intensively than spherical ones [137]. This can be explained by the larger area of contact of flat particles with the cytoplasmic membrane compared to spherical ones. Exocytosis, on the other hand, is more intensive for spherical particles compared to that of non-spherical ones [137]. Therefore, anisodiametrical particles should be preferred, although most authors used spherical ones (Table 1).

### 3.3. Electric Charge

The sign and density of the surface electric charge are important for both the preparation of cytC-bearing hybrid particles and their in vitro or in vivo application to cancer cells. Since cytC globules are positively charged at pH ≤ 9 [88,89], electrostatic adsorption requires a negatively charged particle surface. For example, plate-like alumosilicate clay mineral particles such as kaolinite or montmorillonite (bentonite) are negatively charged due to isomorphic substitution in their crystal lattice by ions of lower valence. This intraparticle charge is pH-independent; therefore, the clay particles are suitable for adsorption of cytC at different pH, which can be chosen considering the protein isoelectric point. Montmorillonite is more suitable because it is a symmetric layered mineral that allows its separation into monoplates, unlike asymmetric kaolinite, whose packs cannot be split owing to the strong dipole–dipole attraction between the monoplates. On the other hand, particles with a pH-dependent charge could be more suitable when their net charge can be changed in a narrow pH range: from negative at pH 7.4 to positive at pH 5. This could allow retaining the adsorbed cytC globules in the extracellular medium and desorbing them in endosomes where the pH is significantly lower.

The total charge of the composite cytC-carrying particles must be positive since the cytoplasmic membrane (plasmalemma) is negatively charged. Particles with a positive surface charge can be easily absorbed owing to their electrostatic attraction to the membrane. This assumption has been demonstrated in a number of experiments with different types of cells and particles: cancer cells absorb more easily positively charged particles of different chemical nature, such as metals (gold, silver, and supermagnetic), oxides (iron oxide, hydroxyapatite, and silica), liposomes, and polymers (polylactic acids, chitosan, and others), compared to their negatively charged counterparts [138]. The surface charge type (positive or negative) and its value have been determined by experimental conditions or by chemical modification of the particle surface.

For example, a higher positive surface charge of the carrier particles leads to higher uptake in mouse macrophages [139]. In dendritic cells, a positive particle charge significantly increases the uptake efficiency [139]. The optimal size for uptake remains 0.5 µm, but the absorption of even larger particles (around one micrometer) can be significantly improved if the particle surface is positively charged [140]. Hydroxyapatite particles, whose surface charge is modified by the adsorption of positively or negatively charged polymers, show higher uptake by cancer cells when they are positively charged compared to their negatively charged analogs. This phenomenon is attributed to the electrostatic attraction between the positively charged particles and the repulsion of negatively charged ones when approaching the negatively charged cell membrane [141].

It is clear that the surface charge requirement is contradictory: Bare particles must be negatively charged to electrostatically adsorb the positively charged cytC globules, but composite cytC-bearing particles must be positively charged to be adsorbed onto the negatively charged cytoplasmatic membrane as the first step of the process of absorption via phagocytosis. This contradiction can be resolved by super-equivalent adsorption of the protein: upon saturated adsorption, the total charge of cytC-montmorillonite particles becomes positive [87].

It has been reported that phagocytosis is enhanced with increasing positive zeta potential, but particles with low potential are not absorbed. One possible explanation of the last case is the lack of electrostatic attraction to the plasma membrane: in the extracellular medium, the high ionic strength (determined mainly by 0.15 M NaCl) leads to shielding of the surface electrostatic potential by the counterions. Another even more logical explanation is the aggregation of particles with low surface charge. Particle size strongly grows at aggregation, impairing phagocytosis. In addition, the particle shape can also change depending on the type of aggregates. For an optimal particle carrier selection, it is also necessary to know how the cytC adsorption is affected by chemical modifiers of the particle surface, pH, plasma proteins, small molecules, and ions, especially bivalent Ca^2+^ and Mg^2+^.

The results, contrary to these logical observations, have been reported in the literature: In some cases, negatively charged particles are absorbed by biological cells [138]. A probable reason for this is the adsorption of plasma proteins or other components of the culture medium to the particle surface. However, this issue is generally not well understood, because the particle charge depends on a variety of factors: the presence of macromolecules (proteins and charged polymers), peptides, divalent ions, buffers, etc., and also pH and ionic strength of the medium. At low particle concentrations and high concentrations of oppositely charged substances in the cancer cell culture, the particle surface can be recharged [142]. The binding of different electrically charged fluorescent molecules used for visualization of particle entry into cells by various techniques (fluorescence microscopy, fluorescence intensity measurement, flow cytometry, etc.) must also be considered.

### 3.4. Particle Concentration

The creation of composite cytC-carrying particles alone is insufficient to achieve high cytotoxicity in vitro and in vivo. The conditions under which the suspension is applied are also important. They must ensure optimal parameters, some of which are fixed (pH 7.4, 0.15 mol/L NaCl), but others can be optimized. The most crucial is the concentration of the cytC particles. However, the requirements for particle concentration are opposite: on the one hand, it must be high to ensure a maximal ratio of the number of particles in the extracellular medium to the number of cancer cells, but, on the other hand, it must be low enough to avoid aggregation, in which the particle size becomes too large for their uptake by phagocytosis. We found a logarithmic dependence of the dead/live cell ratio on the particle concentration in the culture medium [78], which indicates that the cytotoxicity reaches saturation, probably due to the cancer cells’ limited phagocytic capability (absorbed particles per unit time). Thus, an excessively high cytC particle concentration is undesirable. Instead, the concentration should be optimized to maintain a high particle/cell ratio while avoiding aggregation.

## 4. Future Directions

In order to achieve high cytotoxicity competitive with molecular chemotherapeutics, researchers’ efforts must be focused not only on creating overly sophisticated cytC-bearing composite particles but also on their effectiveness as protein carriers, defined primarily by their ability to carry a maximum number of cytC macromolecules in active form to their intracellular release. In addition, cytC particles should be absorbed by cancer cells as easily as possible, which allows for reaching the cytC intracellular concentration threshold at which the caspase cascade is activated to initiate apoptosis faster. Combining cytC with small amounts of a molecular chemotherapeutic adsorbed on the same particles may also increase cytotoxicity.

## 5. Conclusions

The use of composite (nano)particle carriers of cytC is a promising approach for the treatment of neoplasms with superficial localization, as it ensures the avoidance of non-specificity and high toxicity of molecular chemotherapeutics to healthy cells. The high selectivity and low toxicity of cytC-bearing particles can be reached by two approaches, both based on the induction of apoptosis (a harmless for the macroorganism process of cell suicide) by introducing exogenous cytC in cancer cells. The first one relies on the ability of cancer cells to absorb submicron-sized particles via phagocytosis and the inability of healthy differentiated cells to do the same. The second approach is based on the selective introduction of cytC-bearing nanoparticles by receptor-mediated endocytosis. For this purpose, particles are covered by specific small molecules, which bind to membrane receptors that are overexpressed in some cancer cells. However, the cytotoxicity of the majority of composite cytC-bearing particles of different natures and structures described so far in the literature, as determined in in vitro experiments with cancer cell cultures, is low compared to that of chemotherapeutics; so far, this limits the potential of this otherwise very promising (apoptosis-based) approach to be applied in the clinic. Cytotoxicity may be enhanced by considering physicochemical factors in the synthesis of the composite cytC-carrying (nano)particles and the conditions of administration on cancer cell cultures and body neoplasms. In this review, all articles found describing cytC-bearing particles are included in two tables according to the chronology of their publication up to the end of 2024. The articles describing the composition of different types of particles with high cytotoxicity are critically analyzed from a physicochemical point of view, and, on this basis, our recommendations to authors for further development of this perspective and approach in anticancer therapy are proposed.

## Figures and Tables

**Figure 1 pharmaceutics-17-00305-f001:**
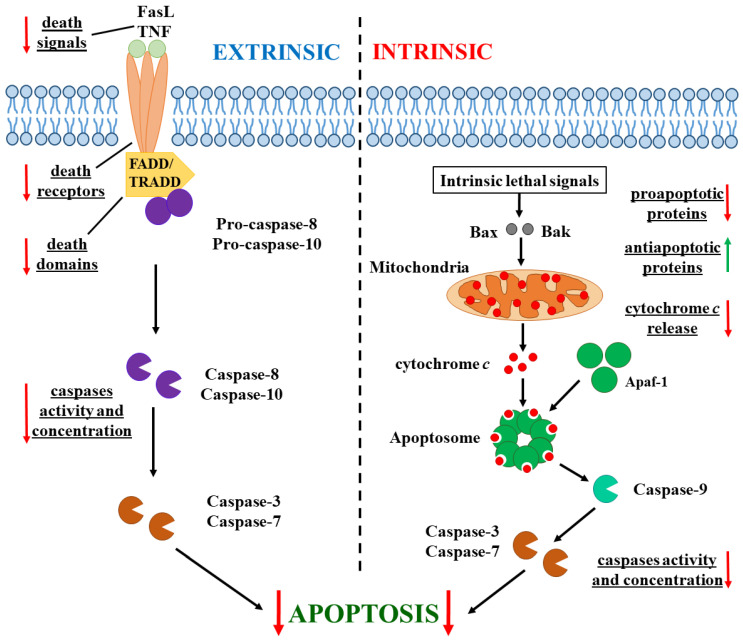
Extrinsic (in the (**left**)) and intrinsic (in the (**right**)) pathways of apoptosis in cancer cells. Alterations (in comparison with apoptosis in normal cells) are indicated by arrows: directed down for decreased stages in the enzyme cascade (the red arrows) and up for increased ones (the green arrow).

**Figure 2 pharmaceutics-17-00305-f002:**
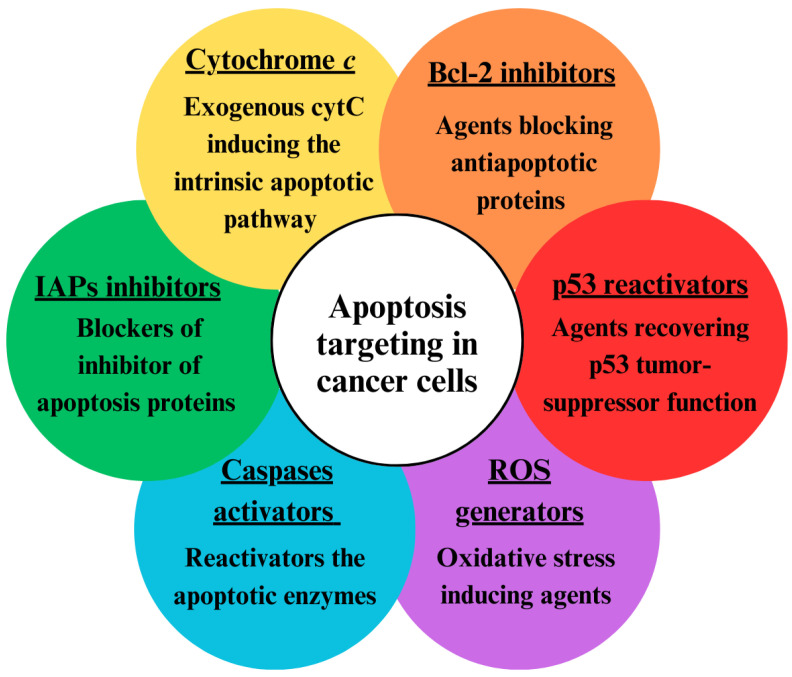
Approaches for targeting apoptosis in tumor cells.

**Figure 3 pharmaceutics-17-00305-f003:**
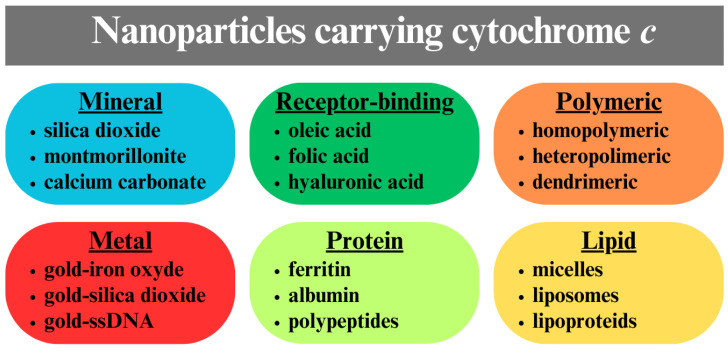
Different types of cytC carrying nanoparticles with a mineral, metal, polymeric, protein, or lipid hybrid core.

**Table 1 pharmaceutics-17-00305-t001:** Physicochemical properties of the composite cytC-carrying nanoparticles (NPs).

	Substance	Shape	Size [nm]	Zeta Potential [mV]	Type Connection	Year	Reference
1	Silica (SiO_2_)	spherical	265	–	noncovalent	2007	[53]
2	Hyperbranched polyhydroxyl polymer	spherical	63 ± 2	–	noncovalent	2010	[54]
3	Apolipoprotein	spherical	20–30	+2–3	–	2012	[55]
4	Antennapedia (peptide)	–	–	–	covalent	2013	[56]
5	Streptavidin, Biotin	spherical	7.06	–	covalent	2013	[57]
6	Poly(lactic-co-glycolic) acid	spherical	100–300	–	covalent	2014	[58]
7	Iron oxide and Gold	spherical	115–200	+10.0	covalent	2014	[59,60]
8	Silica (SiO_2_)	spherical	160–170	–	covalent	2014	[61]
9	Silica (SiO_2_)	spherical	117 ± 10	–14.4 (pH 7.4) –0.5 (pH 5)	–	2014	[62]
10	Silica (SiO_2_)	spherical	356 ± 56	–	covalent	2014	[63]
11	Galactosylated albumin	spherical	13.6 ± 2	–	covalent	2014	[64]
12	Calcium carbonate, PEG	spherical	200–250	−13.7	noncovalent	2015	[65]
13	cytC, Oleic acid	spherical	122.9 ± 5.5	−0.489 ± 3.2	noncovalent	2015	[66]
14	cytC, Folic acid	spherical	338 ± 8	+47.5	covalent	2016	[67]
15	Cardiolipin	spherical	93.8 ± 1.35	−61.1 ± 1.2	noncovalent	2017	[68]
16	cytC, Hyaluronic acid	spherical	542 ± 9	−28.7 ± 0.6	covalent	2017	[69]
17	Iron oxide, Gold	spherical	115–200	+10.0	covalent	2018	[70]
18	Silica (SiO_2_)	spherical	163.3 ± 26.84	–	covalent	2018	[71]
19	Ferritin	spherical	–	–	covalent	2019	[72]
20	Gold, ssDNA	spherical	12 (pH 7.4) 600 (pH 5.5)	−25 (pH 7.4) +3 (pH 5.5)	noncovalent	2019	[73]
21	Silica (SiO_2_)	spherical	73.7 ± 9.2	−16.3 ± 1.0	–	2019	[74]
22	Montmorillonite	monolayer	430	+20.0 (pH 6)	noncovalent	2019 2021	[75,76,77,78]
23	cytC, Folic acid	spherical	253 ± 55 354 ± 11	+26.9 ± 5.03 +22.4 ± 6.36	covalent	2020	[79]
24	Methoxy-PEG-block- -copolymer	spherical	96.3	+15	–	2020	[80]
25	Hyaluronic acid, Triphenylphosphonium	spherical	140	−24.65	covalent	2020	[81]
26	Silica, DMMA, Gold	spherical	200	−15.4 ± 0.3	noncovalent	2021	[82]
27	cytC, Folic acid	spherical	169 ± 9	+17.7 ± 1.7	covalent	2022	[83]
28	PEG, Hyaluronic acid	spherical	50	−10	noncovalent	2022	[84]
29	Cationic dextrin	spherical	100 ± 30	+6.3	noncovalent	2024	[85]

**Table 2 pharmaceutics-17-00305-t002:** Cytotoxicity of the composite cytC-carrying nanoparticles (NPs).

	Nanoparticles	Cancer		Concentration [µg/mL]	Additional Agent(s)	Cytotoxicity (72nd h)	Reference
		Type	Cell Line				
1	Mesoporous silica	Cervical cancer	HeLa	–	–	–	[53]
2	Polymeric	Breast cancer	MCF-7	3200	–	≈65%	[54]
3	Lipid	Lung carcinoma	H460	–	–	in vivo	[55]
4	Peptide	Cervical cancer	HeLa	>1.3	–	clonogenic potential block	[56]
5	Dendritic multidomain	Lung adenocarcinoma	A549	95	–	≈60% ^1^	[57]
6	cytC-polymer	Cervical cancer	HeLa	100	–	≈75–80% ^2^	[58]
7	Hybrid iron oxide–gold	Liver cancer Hepatocellular carcinoma	HepG2	25	–	≈59%	[59,60]
8	Mesoporous silica	Hepatocellular carcinoma	HL-7702 HepG2	–	–	≈80% ^1^	[61]
9	Mesoporous silica	Breast cancer	MCF-7	–	doxorubicin	in vivo	[62]
10	Mesoporous silica	Cervical cancer	HeLa	37.5	Lactose	≈55%	[63]
11	Galactosylated albumin	Liver cancer Hepatocellular carcinoma	HepG2 Hep3B	10	–	≈35% ^1^	[64]
12	CaCO_3_	Breast cancer	MCF-7	50	–	≈35% ^3^	[65]
13	cytC	Cervical cancer Lung adenocarcinoma	HeLa A549	120	Oleic acid	≈95% ^2^ ≈90% ^2^	[66]
14	cytC	Cervical cancer	HeLa	50	Folate	80% ^2^	[67]
15	Cardiolipin	Ovarian cancer	A2780	1	–	≈70%	[68]
16	cytC	Lung adenocarcinoma	A549	164	Hyaluronic acid	≈90% ^2^	[69]
17	Hybrid iron oxide–gold	Liver cancers	HepG2 Huh-7D SK-hep-1	12	doxorubicinpaclitaxel oxaliplatin vinblastine vincristine	≈60–75% ^4^ (maximal with vinblastine)	[70]
18	Mesoporous silica	Hepatocellular carcinoma	HepG2	150	doxorubicin	≈80% ^1^	[71]
19	Ferritin	Acute promyelocytic leukemia (APL)	NB4	150	–	–	[72]
20	Gold-ssDNA	Melanoma (skin)	B16F10	20 nM	–	≈75–80%	[73]
21	Mesoporous silica NPs (degraded)	Ovarian cancer	SKOV3	40	–	≈80% ^1^	[74]
22	Montmorillonite nanoplates	Colon cancer	CCL-227	3	–	97%	[75,76,77,78]
23	cytC	Cervical cancer Lewis lung carcinoma	HeLa LLC	180	Folate	≈85% ^2^ ≈80% ^2^	[79]
24	Polymeric nanomicelles	Liver cancer Hepatocellular carcinoma	HepG2	100	–	≈75% ^1^	[80]
25	Polymeric nanomicelles	In vivo mice Lung adenocarcinoma	A549	50	Paclitaxel	95%	[81]
26	Gold–silica	Mice body tumor	HeLa	50	–	≈80%	[82]
27	cytC	Lewis lung carcinoma	LLC	300	Folate	≈90%	[83]
28	PEG, Hyaluronic acid nanogels	Lung adenocarcinoma	A549	100	–	≈70% ^1^	[84]
29	Cationic dextrin	Cervical cancer	HeLa	1500	chloroquine	≈88% ^5^	[85]

At the ^1^ 24th h; ^2^ 6th h; ^3^ 12th h; ^4^ 48th h, given only for HepG2 cell line; ^5^ 18th h.

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
