# Peer review of "Anticancer Nanoparticle Carriers of the Proapoptotic Protein Cytochrome c"

_pharmaceutics, 2025, doi:10.3390/pharmaceutics17030305_

Round 1
Reviewer 1 Report
Comments and Suggestions for Authors
This review explores how composite nanoparticles carrying the mitochondrial protein cytochrome c can be used in cancer treatment. The review also highlights methods for making these nanoparticles, their properties, and ways to enhance their effectiveness. The comment for revision is as follows
· Conclusive statement should be included in the abstract
· Grammer needs to check e.g. Line 73: For now, however, according………
· Ref number is not as per journal style at present it roman and some where number like line 331 [74], line 388 [88]. Look like authors have not taken the author guidelines seriously. Line 392: [60], [71] line 394: [61, 86, xcii])….its very difficult to trace the work and map with references.
· 2. Particles-carriers of exogenous cytochrome c, here heading should be Nanoparticle…… in place of particle
· Line 423: Avoid using we, them, I etc.
· Line 298 to 329, research study text needs to cite. Please check full manuscript for such error where due acknowledgement given.
· The paragraph and section need to connect, and flow should be improve.
· Why Montmorillonite nanoplates, are there where we are discussing nanoparticles?
· Line: In our opinion, the main drawback of this approach is that the CD71 receptor, such statement need support as it not research and not data is there for stating such a thing.
. Table 2, needs to be updated with work on other nanoparticles like Platinum–Iron Nanoparticles (https://doi.org/10.3390/inorganics12120331), polymeric micelles (https://doi.org/10.1016/j.colsurfb.2024.113871) (Metal-Organic Frameworks
· Line 948: Indian authors (Sarkar at al.), year should be there and not at al. it must be et al.
· Line 1185: As we noted in Section 2.9, the use of porous particles, however, 2.9. is about Cationic dextrin,
· In 2.9 many texts look irrelevant please delete it.
. 1.1. Anticancer strategies, can be update.
1.2 Apoptosis: Discuss in brief about recent trends and work like apoptotic bodies derived from 2D- and 3D- cultured stem cells
· Overall, the manuscript needs to be rewritten and reduced the length with excessive research data elaboration at present. Please stick to the main topic and improve the flow and connection between sections and paragraphs.
· Suggestion to checking language carefully in revised version.
Reviewer 2 Report
Comments and Suggestions for Authors
ANTICANCER NANOPARTICLES-CARRIERS OF THE PROAPOPTOTIC PROTEIN 2 CYTOCHROME C
ü How selectively the apoptosis can occur in the cancer cells
ü In Au-cytC-SiO2, Au nanoparicles completely plugged the pores and how to ensure that?
ü What is ideal size of nanoparticles?
ü Which gives the better selectivity Au or Fe3O4?
ü Page no. 13 spelling mistake highlighted in yellow color
ü What are drawbacks of microinjection?
ü These nano carriers are suitable for all type of cancer for selectivity?
ü In form of nano composite, CytC is how long stable?
ü Any ways to change the surface electric potential of the pores to enhance the adsorption
ü There is no evidence mentioned for invivo studies of nanocarries selectivity
ü How the shapes will affect the adsorption and cytotoxicity?
ü Among the nano carriers mentioned which has better cytotoxicity, selectivity and cancer cell apoptosis?
Reference are represent is roman letters and numerical. Reference is not uniform –not cleared .
Comments on the Quality of English Languageneeds improvement
Reviewer 3 Report
Comments and Suggestions for Authors
The manuscript by Zhivkov et al. highlights composite nanoparticles (NPs) carrying cytochrome c (CytC), a mitochondrial protein that triggers apoptosis when introduced into cells via phagocytosis. These NPs offer a controlled delivery system for CytC, presenting a promising approach for treating superficial cancers by ensuring selectivity. The review also explores factors influencing cytotoxicity, enhancement strategies, and NP preparation methods. The authors have several papers in the field, and this review could be valuable to the field, especially given the lack of recent reviews on this topic. However, I believe there are serious issues that must be addressed to improve the manuscript's quality. Below are my comments:
Major Comments
Language and Writing Quality:
The manuscript is poorly written and requires significant revision to improve its language and clarity.
Several phrases and pieces of information are incorrect or unclear, likely due to language issues. For example:
“Traditional chemotherapeutics are administrated systematically.”
“Since only cancer cells (with the exception of cells of the immune system – neutrophils and monocytes mainly) have the ability to phagocytose, so apoptosis can be induced only in cancer cells but not in healthy one.”
These statements need to be corrected for accuracy and clarity.
Introduction:
The introduction is poorly structured and lacks sufficient detail. It needs to provide accurate and comprehensive background information.
The current format appears oversimplified, as if written for a general audience rather than a scientific readership.
Use of First Person:
Many sections are written in the first person, which is not appropriate for a scientific review. Examples include:
“We believe…”
“Our assessment of this work is…”
These phrases should be revised to maintain a formal and objective tone.
Referencing and Consistency:
The referencing style is inconsistent throughout the manuscript, particularly in the text and tables. This needs to be standardized.
Flow and Structure:
The manuscript lacks coherence and consistency in its arguments and presentation. The text should be reorganized to ensure a logical flow of ideas.
Minor Comments
Abbreviations:
Abbreviations must be used consistently throughout the manuscript. Ensure all abbreviations are defined upon first use.
Illustrations and Graphs:
Including illustrations or graphs would make the manuscript more informative and engaging for readers.
Addressing the issues outlined above would significantly enhance the quality of the manuscript. I hope these suggestions are helpful in improving the clarity, accuracy, and overall impact of the review.

The manuscript is poorly written and requires significant revision to improve its language and clarity.
Round 2
Reviewer 1 Report
Comments and Suggestions for Authors
Revision is acceptable
Reviewer 2 Report
Comments and Suggestions for Authors
revisions are satisfactorily carried out
Reviewer 3 Report
Comments and Suggestions for Authors
Your revised manuscript is of high quality and makes a significant contribution to the literature. The revisions have strengthened the clarity, rigor, and impact of your work, making it a valuable resource for researchers in the field.